# A stress assembly that confers cell viability by preserving ERES components during amino-acid starvation

Margarita Zacharogianni[1,2], Angelica Aguilera Gomez[1,2], Tineke Veenendaal[1,2,3], Jan Smout[1,2], Catherine Rabouille[1,2,3]*

[1]Hubrecht Institute, Royal Netherlands Academy of Arts and Sciences, Utrecht, Netherlands; [2]University Medical Center Utrecht, Utrecht, Netherlands; [3]Department of Cell Biology, University Medical Center Utrecht, Utrecht, Netherlands

**Abstract** Nutritional restriction leads to protein translation attenuation that results in the storage and degradation of free mRNAs in cytoplasmic assemblies. In this study, we show in Drosophila S2 cells that amino-acid starvation also leads to the inhibition of another major anabolic pathway, the protein transport through the secretory pathway, and to the formation of a novel reversible non-membrane bound stress assembly, the Sec body that incorporates components of the ER exit sites. Sec body formation does not depend on membrane traffic in the early secretory pathway, yet requires both Sec23 and Sec24AB. Sec bodies have liquid droplet-like properties, and they act as a protective reservoir for ERES components to rebuild a functional secretory pathway after re-addition of amino-acids acting as a part of a survival mechanism. Taken together, we propose that the formation of these structures is a novel stress response mechanism to provide cell viability during and after nutrient stress.

*For correspondence:
c.rabouille@hubrecht.eu

**Competing interests:** The authors declare that no competing interests exist.

**Reviewing editor**: Jodi Nunnari, University of California, Davis, United States

## Introduction

Cell response to nutritional restriction includes stimulation of degradation pathways, such as autophagy, as well as attenuating anabolic pathways, such as protein translation (*Castilho et al., 2014*).

Another key anabolic pathway is protein transport through the secretory pathway. In mammals, one third of the proteome encounters this pathway (*Stevens and Arkin, 2000*; *Almen et al., 2009*), such as the proteins delivered to the extracellular medium, the plasma membrane, or other cellular membrane compartments with the exception of mitochondria and the nucleus. After their synthesis at the endoplasmic reticulum, proteins exit the ER at specialized ER exit sites (ERES) defined by a cup-shaped ER overlaying COPII-coated vesicles in which newly synthesized proteins are packaged. They then reach the Golgi apparatus where they are further modified, sorted, and dispatched to their correct final localization. The COPII coat assembly requires 6 proteins, including the transmembrane protein Sec12 that acts as a GEF for the small GTPase Sar1. GTP-bound Sar1 recruits Sec23/Sec24, the inner COPII coat that in turn recruits Sec13/31, the outer coat (*Bard et al., 2006*; *d'Enfert et al., 1991*; *Oka et al., 1991*; *Rothman and Wieland, 1996*; *Schekman and Orci, 1996*). In addition, the large hydrophilic protein Sec16 has been found to play a major role in the COPII assembly and regulation (*Ivan et al., 2008*; *Hughes et al., 2009*) (*Connerly et al., 2005*; *Kung et al., 2012*; *Bharucha et al., 2013*) and Sec16 mutation or loss of function leads to a severe impairment in trafficking through the secretory pathway.

Stress strongly affects the functional organization of the secretory pathway. For instance, energy deprivation and osmotic shock also block secretion at the level of the ER exit and the cis Golgi, a response mostly triggered by impaired dynamics of the COPI coat, which mediates retrograde

**eLife digest** Proteins are needed by living cells to perform vital tasks and are made from building blocks called amino-acids. However, if a cell is starved of amino-acids, protein assembly comes to a halt, and if cells are deprived of amino acids for a long time, the cell may die.

To survive short periods of amino-acid starvation, the cell has developed many protective mechanisms. For example, it can start to break down existing proteins, allowing the cell to scavenge and reuse the amino-acids to make other proteins that are more important for short-term survival. The cell may also temporarily halt certain processes: for example, newly constructed proteins may no longer be transported from the cell structure where they are made—called the endoplasmic reticulum—to their final destinations in the cell. However, the protein transport apparatus is also made of proteins and needs to be protected from being broken down so that once starvation ends, the cell can more quickly return to normal working order.

Zacharogianni et al. identify a strategy cells use to store and protect part of their protein transport apparatus during times of stress. Starving fruit fly cells of amino-acids causes the cells to form protective stress assemblies incorporating the proteins associated with the 'exit sites' that release proteins from the endoplasmic reticulum. These assemblies are called Sec bodies, and when amino-acid starvation ends, these bodies release the exit site components unharmed. This allows the cell to quickly resume protein transport and so speeds the cell's recovery. If the Sec bodies do not form, the cells are more likely to die during amino-acid starvation.

The Sec bodies are distinct from previously identified stress assemblies that form in the cell during stress, but they share features with them, such as being liquid droplets. Some of these assemblies have been linked to degenerative diseases like amyotrophic lateral sclerosis (ALS). Further research will be necessary to determine if there are any similar harmful side effects associated with the formation of Sec bodies.

transport (*Jamieson and Palade, 1968*; *Cluett et al., 1993*; *Lee and Linstedt, 1999*). Interestingly, GBF1, the GEF of Arf, the small GTPase required for COPI assembly, is phosphorylated and consequently inactivated by AMPK under conditions of nutrient starvation and energy depletion, leading to a block in secretion (*Miyamoto et al., 2008*). Furthermore, biosynthesis of PI4P in yeast that was shown to play a key role in coordinating trafficking from the Golgi with cell growth seems to also be sensitive to nutrient conditions (*Piao et al., 2012*). Last, ER stress that elicits the so-called 'unfolded protein response' (*Shamu et al., 1994*) directly impedes on the functional organization of ERES in Drosophila S2 cells (*Kondylis et al., 2011*) and reduces COPII subunits assembly in human cells (*Amodio et al., 2009*). Furthermore, we have recently reported that serum starvation of Drosophila S2 cells also results in a distinct change in the ERES organization, namely Sec16 cytoplasmic dispersion away from ERES, in a conserved ERK7-dependent mechanism (*Zacharogianni et al., 2011*) that leads to protein secretion inhibition.

In this study, we focus on amino-acid starvation that leads to the formation of a novel, non-membrane bound cytoplasmic stress assembly that contains ERES components and that we call 'Sec bodies' [*Figure 1A* and (*Zacharogianni et al., 2011*)]. Sec bodies do not represent terminal aggregates. They are reversible, act as a reservoir for ERES components to reconstruct a functional secretory pathway upon re-feeding and are critical for cell survival during stress and upon stress relief. Furthermore, they display properties similar to those of Stress Granules, which place them in the rapidly growing class of cytoplasmic mesoscale assemblies and more specifically, the category of liquid droplets.

## Results

### Amino-acid starvation of Drosophila S2 cells induces the remodeling of ERES components into Sec bodies

Amino-acid starvation of Drosophila S2 cells (i.e., cell incubation in Krebs Ringers Bicarbonate buffer, KRB, (*Gaccioli et al., 2006*)) leads to inhibition of protein transport through the secretory pathway, as shown by monitoring the plasma membrane localization of the transmembrane reporter Delta

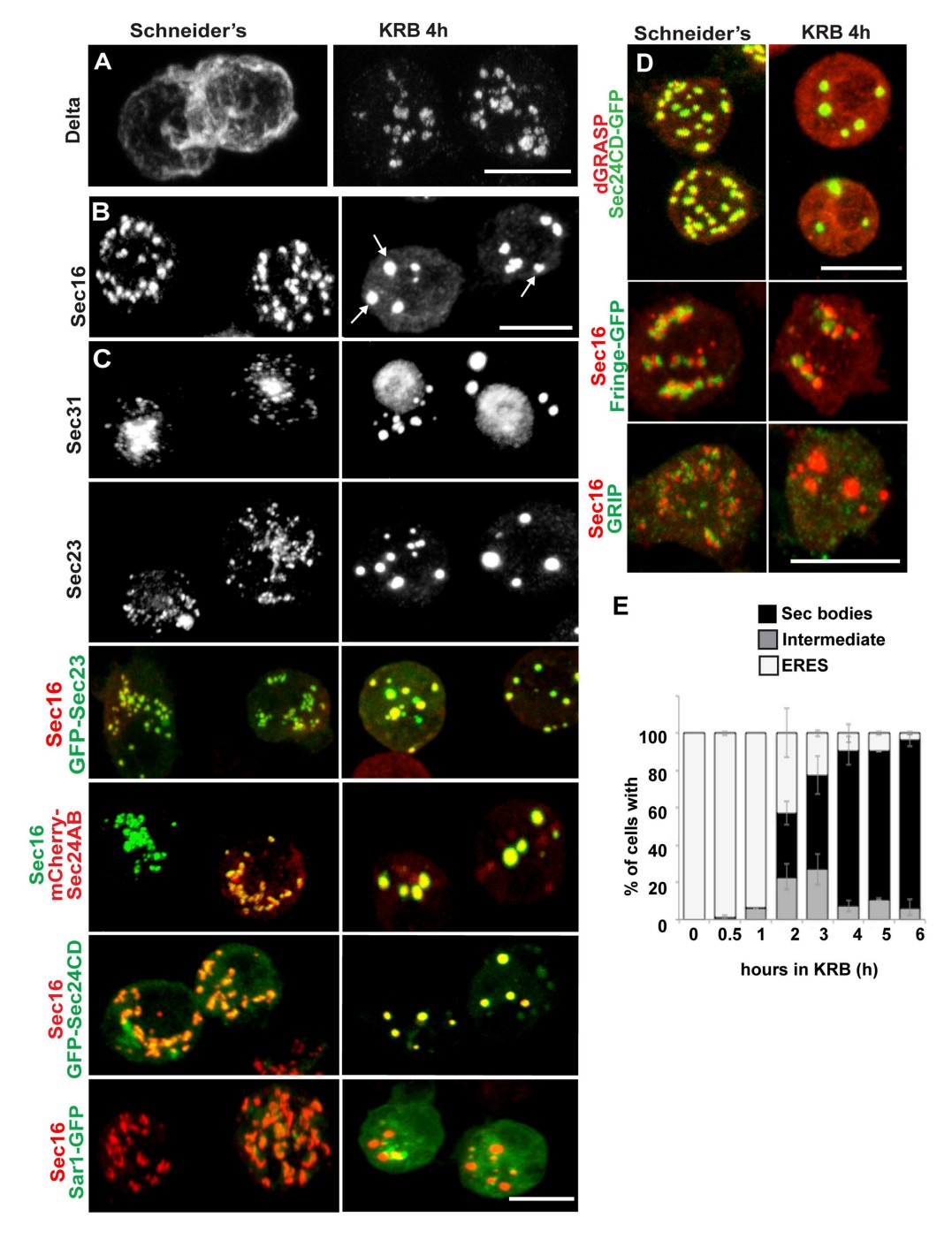

**Figure 1**. Amino-acid starvation induces the formation of a novel stress assembly in Drosophila S2 cells.
(**A**) Immunofluorescence (IF) visualization of Delta-myc (using an anti-Delta antibody) in S2 cells in Schneider's (normal growth conditions) or incubated with Krebs Ringer Bicarbonate buffer (KRB) for 4 hr (amino-acid starvation). Note that in Schneider's Delta reaches the plasma membrane whereas it is retained intracellularly in starved cells. (**B**) IF visualization of endogenous Sec16 in Drosophila S2 cells grown in Schneider's and incubated in KRB for 4 hr. Note the formation of Sec bodies (arrows). (**C**) IF visualization of Sec31, Sec23 and co-visualization of GFP-Sec23, mCherry-Sec24AB, Sec24CD-GFP, Sar1-GFP with Sec16 in S2 cells in Schneider's and KRB for 4 hr. (**D**) IF co-visualization of dGRASP/Sec24CD-GFP, GRIP/Sec16, and Fringe-GFP/Sec16 in S2 cells grown in Schneider's and incubated in KRB for 4 hr. (**E**) Kinetics of Sec body formation in S2 cells incubated in KRB over indicated time
*Figure 1. Continued on next page*

*Figure 1. Continued*

(up to 6 hr) expressed as the percentage of cells exhibiting ERES, intermediates (see 'Materials and methods'), and Sec bodies. Scale bars: 10 μm (**A**–**D**).

The following figure supplements are available for figure 1:

**Figure supplement 1**. Sec body formation and autophagy.

**Figure supplement 2**. Sec body formation and single amino-acids.

**Figure supplement 3**. Sec body formation in vivo.

**Figure supplement 4**. Sec body formation and mammalian cells.

**Figure supplement 5**. Human Sec16A is incorporated to Sec bodies in starved Drosophila S2 cells.

(*Kondylis and Rabouille, 2003*). In cells grown in Schneider's, Delta reaches the plasma membrane, whereas in KRB, it is retained intracellularly (*Figure 1A*). Amino-acid starvation also results in the formation of novel Sec16-positive spherical structures (*Figure 1B*). In addition to Sec16, these structures also contain COPII subunits Sec23, the two Sec24 orthologs Sec24AB (CG1472, Hau) (*Norum et al., 2010*) and Sec24CD (CG10882, Sten) (*Forster et al., 2010*), and Sec31, and we therefore name them 'Sec bodies'. Conversely, Sec bodies do not contain Sar1 (*Figure 1C*), COPI components and clathrin (not shown). They also do not contain dGRASP (that amino-acid starvation drives to complete dispersion in the cytoplasm), the TGN GRIP-domain protein dGCC185, and the Golgi integral membrane protein, Fringe-GFP (*Figure 1D*), although these two latter proteins are often found in close proximity to Sec bodies. The morphology of the ER, on the other hand, does not seem affected by amino-acid starvation (not shown).

Quantitation of this starvation phenotype reveals that although Sec bodies are present in 20% of cells after 1 hr of amino-acid starvation, it takes between 4 hr and 6 hr to get 90% of the cells displaying the typical Sec body pattern (*Figure 1E*), that is 7 ± 3 Sec bodies/cell, including 1 to 5 with a diameter comprised between 0.6 and 0.8 μm. Interestingly, 4–6 hr corresponds to the end of the autophagy peak, a degradative pathway stimulated by starvation (*Klionsky et al., 2012*) (*Figure 1—figure supplement 1A'*) as assessed by the appearance of Atg5 punctae (*Figure 1—figure supplement 1A,A'*). In this regard, pharmacological inhibition of autophagy (by wortmanin or bafilomycin) results in a premature formation of Sec bodies (*Figure 1—figure supplement 1B,B'*), consistent with the notion that Sec bodies form in response to a reduced level of intracellular amino-acid concentration.

Given the Sec body content in COPII subunits, we used immuno-electron microscopy (IEM) to test whether they are not simply a collection of COPII vesicles. Sec bodies are electron dense structures and non-membrane bound, although small membrane profiles can occasionally be observed in their core and often ER is in close proximity (*Figure 2A*, arrows). Sec body formation is associated with the loss of the typical early secretory pathway morphology (*Kondylis and Rabouille, 2009*). ERES and Golgi stacks are no longer visible.

Sec body formation is specific for amino-acid starvation as heat shock, ER stress (tunicamycin and DTT treatment), glucose starvation, oxidative stress (arsenate treatment), hypoxia (1% $O_2$ for 19 hr), and respiration uncoupling (CCCP for 4 hr) do not lead to this response (not shown). Furthermore, to assess whether Sec bodies form in response to the withdrawal of specific amino-acids, cells were incubated in KRB in the presence of individual amino-acids. Histidine, aspartate, and asparagine (at 15 mM) strongly prevent Sec body formation but others also do so, albeit more mildly (*Figure 1—figure supplement 2*), suggesting that the signaling pathway is complex. Taken together, we demonstrate that amino-acid starvation leads to the remodeling of ERES components into Sec bodies.

Sec bodies seemingly also form in vivo, for instance in ovaries starved either ex vivo or dissected from starved female flies (*Figure 1—figure supplement 3*). We also asked whether they form in mammalian cells. Although we do observe some degrees of remodeling of COPII components upon mammalian cell starvation, it remains unclear whether the resulting structures are Sec bodies

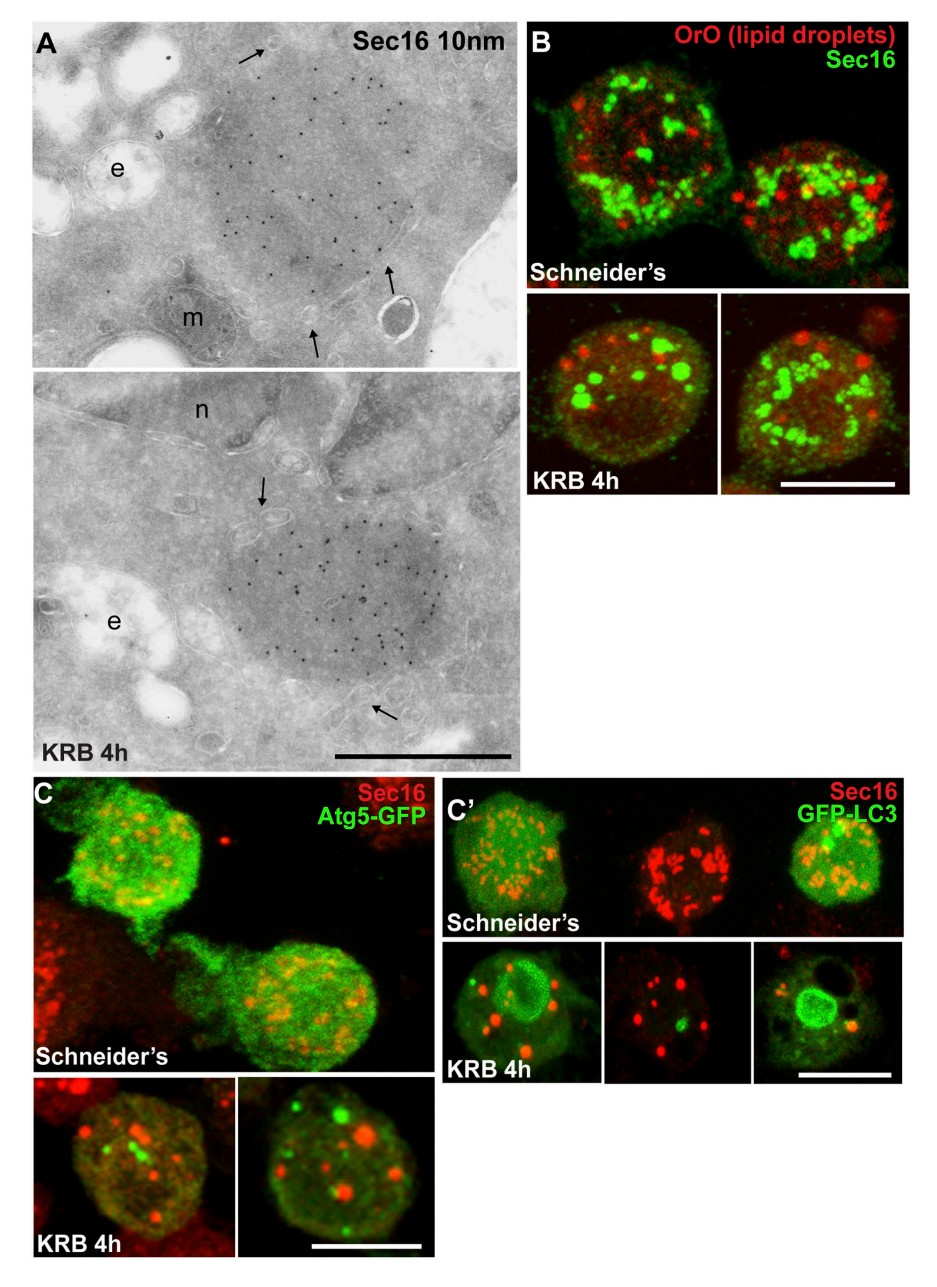

**Figure 2**. Sec bodies are non-membrane bound structures. (**A**) Immuno-electron microscopy (IEM) visualization of Sec16 (10 nm colloidal gold) in Sec bodies in ultrathin sections of S2 cells incubated in KRB for 4 hr. Arrows point to membrane in close proximity of Sec bodies. E, endosomes; n, nucleus; m, mitochondria. (**B**) Visualization of Sec bodies (Sec16, green) and lipid droplets (marked by oil-red-O, red). Note that 95% of Sec bodies do not co-localize with lipid droplets. (**C–C'**) Visualization of Sec bodies (Sec16, red) and Atg5-GFP punctae (**C**) and GFP-Atg8 (**C'**) after 4 hr starvation. Note that 82% of Sec bodies do not co-localize with Atg5 or Atg8 punctae. Scale bars: 500 nm (**A**); 10 μm (**B**, **C**).

(*Figure 1—figure supplement 4*). However, when human Sec16A is transfected in S2 cells, it is efficiently recruited to Sec bodies along with the endogenous components (*Figure 1—figure supplement 5*). This suggests that in mammalian cells, Sec body formation is perhaps less pronounced due to different signaling events and not to the properties of the ERES components themselves (at least Sec16).

## Sec bodies are a novel stress assembly

The IEM analysis shows that Sec bodies are not endosomes or lipid droplets, as their ultrastructure is very different from these organelles [see (*Teixeira et al., 2003*), for lipid droplet ultrastructure in Drosophila]. This is confirmed by the fact that Sec bodies are negative for neutral lipids stained by oil red O that stains lipid droplet content (*Figure 2B*).

As mentioned above, amino-acid starvation is known to induce autophagy, but we demonstrate that Sec bodies are not autophagosomes, as they do not co-localize with Atg5 or Atg8 (*Figure 2C*). Furthermore, as Sec bodies do not contain dGRASP, they are clearly different from the recently described yeast 'compartment for unconventional protein secretion' (CUPS) (*Bruns et al., 2011*).

Amino-acid starvation also results in protein translation inhibition/stalling that leads to the accumulation of untranslated mRNAs. Those are stored in Stress Granules (*Kedersha et al., 1999*; *Anderson and Kedersha, 2008*), or degraded in Processing Bodies (P-Bodies), both cytoplasmic ribonucleoprotein particles (RNPs) comprising mRNAs, RNA binding proteins, RNA processing machineries (P-bodies), and translation initiation factors (Stress Granules). We therefore tested whether Sec bodies are related to these structures. We visualized Stress Granules using endogenous FMR1 (Fragile X mental retardation protein 1), an RNA binding protein, and eIF4E, a translation initiation factor, and P-bodies with Tral (Trailer Hitch), a like-SM protein. In normal growth conditions, FMR1, eIF4E, and Tral are largely diffuse in the cytoplasm and Tral is also found in small punctae representing steady-state P-bodies (*Figure 3A*) (*Eulalio et al., 2007*). Upon amino-acid starvation, as expected, Stress Granules form and P-Bodies enlarge (*Figure 3A*) (*Shimada et al., 2011*) in agreement with reported phenotypes in many cell types, (*Buchan et al., 2008*; *Stoecklin and Kedersha, 2013*). In S2 cells, they form a dual structure, Stress Granule/P-Bodies (SG/PB), in which FMR1 strictly co-localizes with Tral (*Figure 3A*).

Sec bodies and SG/PBs form under the same conditions and in the same time frame, and although they have a spatial relationship and are often found adjacent to each other, Sec bodies are clearly distinct from SG/PBs (*Figure 3B*). To confirm that they are indeed different structures, we also performed IEM of FMR1 and Tral in starved cells (*Figure 3C*) and compared them to Sec bodies (*Figure 2A*). The FMR1/Tral positive structures are less electron dense, round and regular, and they appear to be often surrounded by mitochondria, which is not the case for Sec bodies.

Taken together, these results show that Sec bodies are a novel stress assembly triggered by amino-acid starvation that is distinct from compartments and structures that are also formed upon this condition.

## Sec bodies form at ERES

We then asked how Sec bodies form. Time-lapse imaging of live cells using GFP-Sec23 reveals that once the cells sense amino-acid depletion, a pool of ERES rapidly disappears by releasing their components in the cytoplasm, and the remaining ones are rapidly transformed into smaller round structures. These small structures do not seem to efficiently fuse with one another. Instead, they seem to act as a seed and grow by recruiting ERES components from the cytoplasm where they were released to reach the typical Sec body size (*Figure 4A*, and *Video 1*).

To assess, as suggested by the time-lapse, whether Sec bodies form at ERES, we used a truncated version of Sec16, miniSec16 (690-1954), which comprises the minimal Sec16 sequence required for ERES localization (*Ivan et al., 2008*) (*Figure 4B*) but is not incorporated into Sec bodies upon amino-acid starvation (*Figure 4B'*). Instead, miniSec16 remains associated to the cup-shaped ER of the ERES and seems to cradle the forming Sec bodies (marked by endogenous Sec16; *Figure 4B'*). This indicates that Sec bodies form where ERES were present, in line with their observed proximity to ER membrane (arrows in *Figure 2A*).

The time-lapse also suggested that the Sec body enlargement is mediated by recruitment of ERES components that have been dispersed in the cytoplasm. To test this further, we used a Sec16 deletion mutant (ΔNC2-3) that does not localize to ERES and is mostly cytoplasmic (*Figure 4B*) because it lacks the region that mediates its recruitment to ERES (NC2-3) (*Ivan et al., 2008*). We found that it is efficiently recruited to Sec bodies, showing that Sec16 can be recruited from the cytoplasm and contributes to Sec body enlargement (*Figure 4B"*). Furthermore, this result indicates that the localization to ERES prior to starvation is not necessary for incorporation to Sec bodies. Importantly, the recruitment of this cytosolic mutant is not due to its interaction with endogenous Sec16 as the NC2-3 also contains the Sec16 oligomerization domain (*Ivan et al., 2008*). This suggests that a distinct Sec16 domain responds to amino-acid starvation.

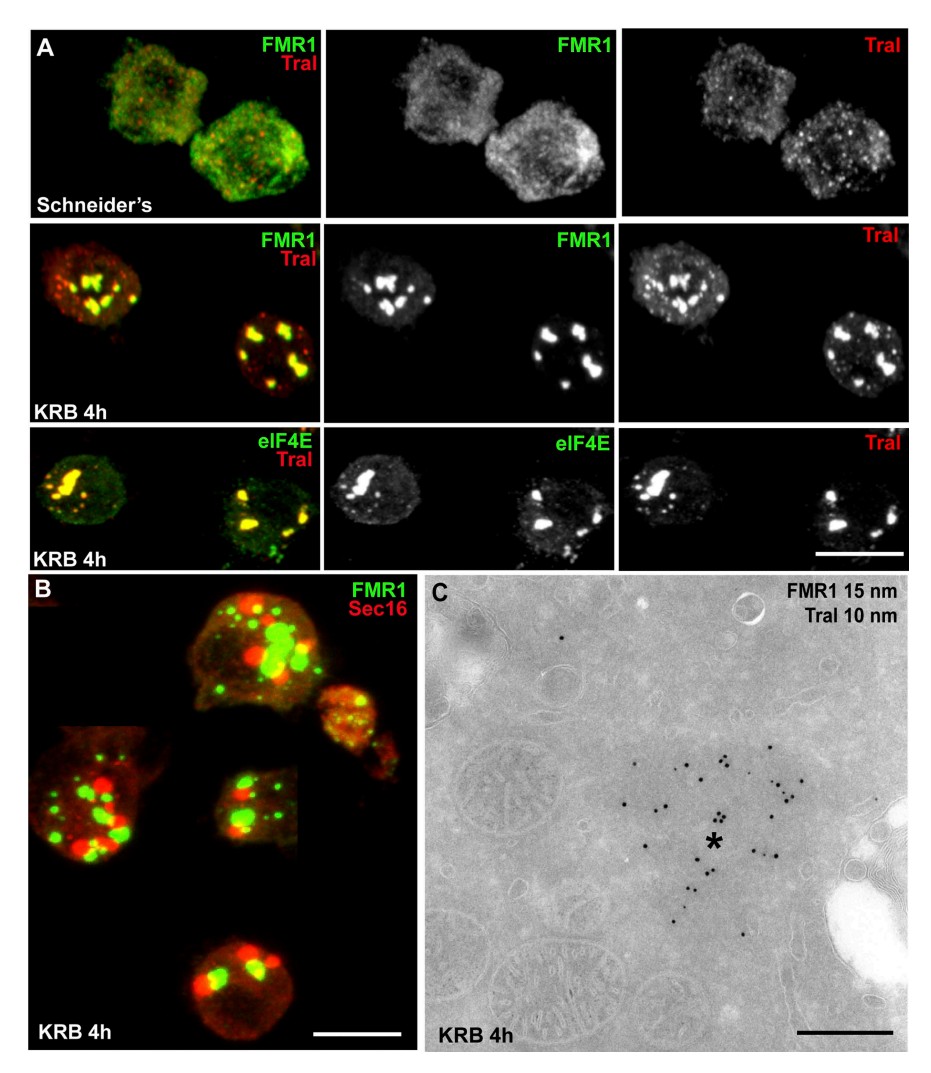

**Figure 3**. Sec bodies are distinct from Stress Granules and P-bodies. (**A**) IF visualization of endogenous FMR1, eIF4E (green), and Tral (red) in cells growing in Schneider's and incubated in KRB for 4 hr. Note that upon starvation, Stress Granules (FMR1, eIF4E) form and P-Bodies (Tral) enlarge to co-localize in SG/PBs. (**B**) IF visualization of Sec16 (Sec bodies) and FMR1 (SG/PB) in cells incubated in KRB for 4 hr. Sec bodies and SG/PBs are distinct structures but have a spatial relationship. (**C**) IEM visualization of FRM1 and Tral in ultrathin sections of S2 cell incubated with KRB for 4 hr. Note that the Tral and FMR1 positive SG/PBs (asterisk) are clearly different from Sec bodies (*Figure 2A*). Scale bars: 10 μm (**A**, **B**); 500 nm (**C**).

Taken together, these results show that Sec bodies form (at least initially) where ERES were located and increase in size by recruiting ERES components dispersed in the cytoplasm.

## Sec body assembly does not require active transport through the early secretory pathway but depends on specific ERES components

We then ask whether membrane traffic through the early secretory pathway is required for Sec body formation. To test if COPII vesicle formation is required, we depleted Sar1 by RNAi before starvation. Sar1 depletion is evidenced by a strong reduction (40 ± 5%) in cell proliferation and ERES enlargement (*Ivan et al., 2008*) (*Figure 4C*, arrows). However, Sec body formation was found to be as efficient as in control (mock depleted) cells (*Figure 4C*). Second, we pharmacologically inhibited protein trafficking with Brefeldin A (*Figure 4D*) and found that this treatment before and during starvation does not affect Sec body formation (*Figure 4E*). This demonstrates that transport in the early secretory

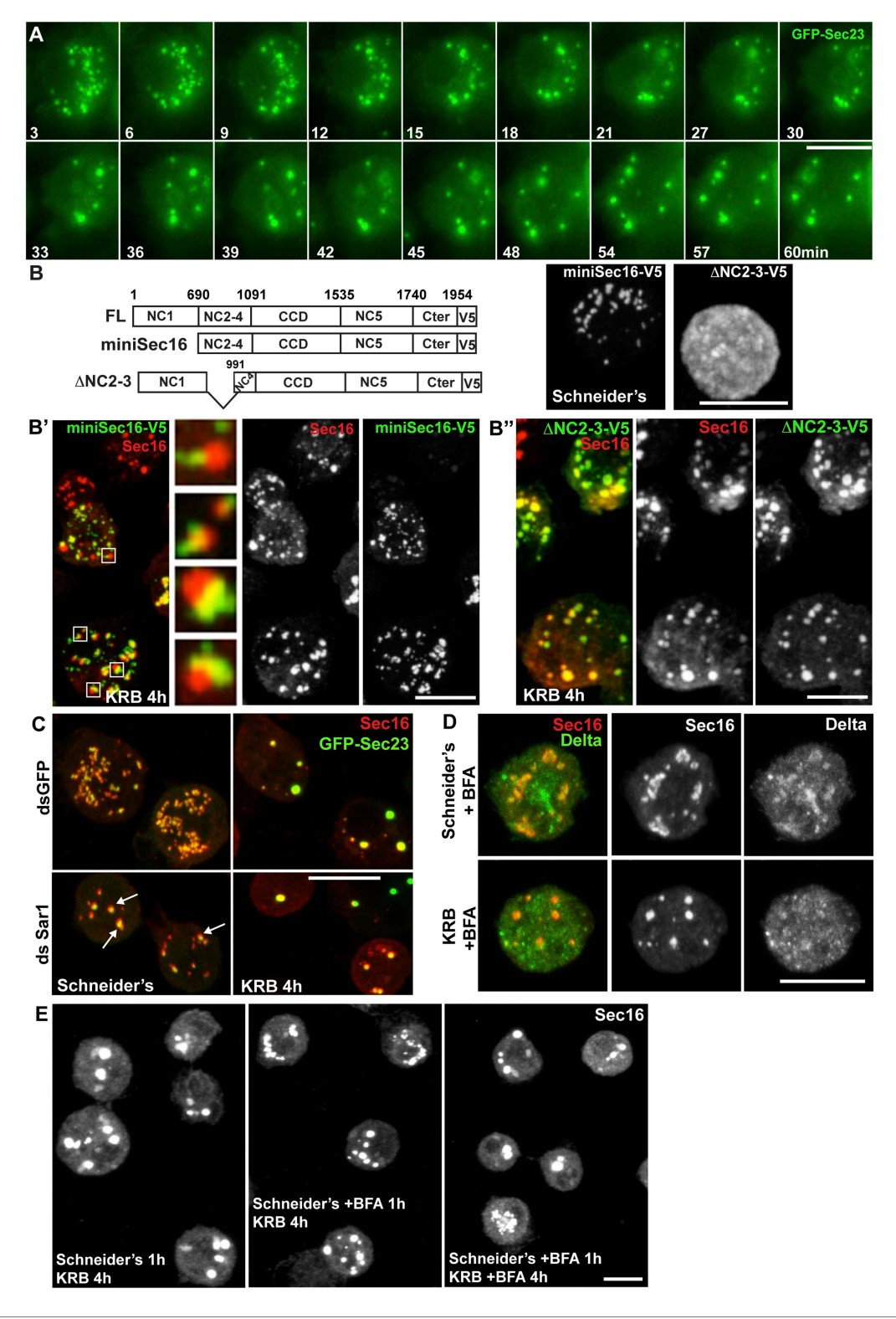

**Figure 4**. Sec bodies form at ERES but COPII- and COPI-coated vesicle formation is not required. (**A**) Stills of a GFP-Sec23 time-lapse video of a cell incubated in KRB (t = 0) for 60 min showing Sec body formation. (**B–B''**) IF visualization of miniSec16-V5 (**B**, **B'**) and ΔNC2-3-Sec16-V5 (**B**, **B''**) in S2 cells incubated in Schneider's (**B**) and KRB for 4 hr. Endogenous Sec16 is in red. Note that the cup-shaped ER forms a cradle for Sec bodies (**B'**, insert).
*Figure 4. Continued on next page*

*Figure 4. Continued*

(**C**) IF visualization of Sec16 and GFP-Sec23 in mock and Sar1-depleted S2 cells grown in Schneider's and incubated in KRB for 4 hr. Note that the ERES are enlarged in Sar1-depleted cells (arrows) and Sec bodies form in both conditions to the same extent. (**D**) IF visualizations of Sec16 and Delta-myc in S2 cells incubated with brefeldin A (BFA) in Schneider's and KRB for 3 hr. Note that Delta transport is inhibited in both cases as Delta is retained intracellularly. (**E**) IF visualization of Sec16 in S2 cells grown in Schneider's and incubated in KRB for 4 hr in the presence or absence of brefeldin A (BFA). Note that pre-incubation with the drug does not affect Sec body formation during starvation. Scale bars: 10 μm.

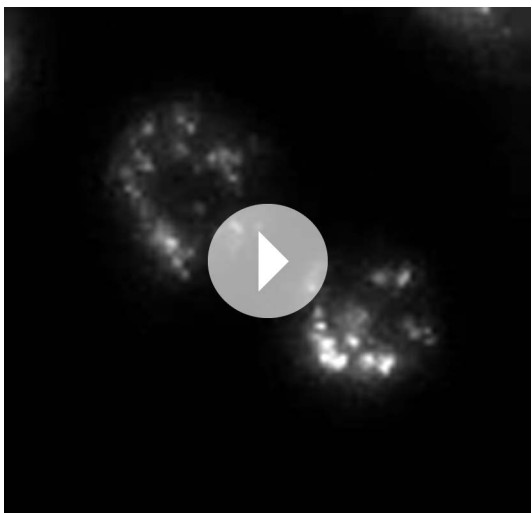

**Video 1**. Time-lapse movie of Sec body formation. GFP-Sec23 time-lapse video of two cells incubated in KRB (t = 0) for 1 hr. One frame was taken every 3 min and the videos are displayed at 7 frame/s (related to **Figure 4A**).

pathway via COPI and COPII vesicle formation is not required for the formation of Sec bodies.

Given the Sec body content in ERES components (**Figure 1**), we then tested using RNAi if they are necessary for Sec body formation. Depleting Sec16 did not yield satisfactory conclusions as we have shown that Sec16 is critical for the organization of ERES in Drosophila S2 cells (**Ivan et al., 2008**) and Sec16 depletion led to the aggregation of most of the COPII components even in cells grown in full medium (see **Figure 2A–C″** of **Ivan et al., 2008**). However, in the few depleted cells where an observation could be made, Sec bodies did not form (not shown).

We then depleted the two Sec24 gene products that are both expressed in S2 cells (see DRSC, Drosophila RNAi screening center, http://www.flyrnai.org/) (see 'Materials and methods' for depletion controls). When Sec24AB-depleted cells are starved, the normal Sec body formation (marked by Sec16, **Figure 5A**) is impaired (**Figure 5B**). The distribution of diameters of the resulting structures shows that they are twofold smaller and twice as many, when compared to Sec bodies in mock-depleted cells (**Figure 5E**). In agreement with Sec24 forming a complex with Sec23, Sec23 depletion also results in the same phenotype (**Figure 5C,E**). These smaller structures are not classical Sec bodies. By IF, some of them appear to have a horseshoe shape. By IEM, they appear as a collection of Sec16 positive vesicular and tubular membrane profiles, some small, some large probably corresponding to ERES mixed with Golgi fragments, and a third category that we name 'intermediates' as they are reminiscent of Sec bodies by their round shape but that contain membrane and are smaller in size when compared to Sec bodies found in mock-depleted cells (**Figure 5—figure supplement 1**).

Interestingly, Sec24CD depletion (**Figure 5D,E**) does not lead to the same phenotype as Sec24AB depletion and Sec bodies form seemingly normally, showing specificity for one Sec24 homologue.

Taken together, this indicates a key and novel role for Sec23, Sec24AB (and perhaps Sec16) in Sec body formation, which is distinct from their classical role in ER exit via COPII vesicle formation that is not required (**Figure 4C–E**).

## Sec bodies display liquid droplet-like properties: role of low complexity sequences

Given that the Sec bodies are non-membrane bound, we then asked how their components are prevented from freely diffusing in the cytoplasm. One class of stress related cytoplasmic structures are liquid droplets that are described to result from phase separation-induced liquid demixing in the cytoplasm (**Brangwynne et al., 2009, 2011**; **Hyman and Simons, 2012**). They are defined as non-membrane bound, spherical, reversible structures, the components of which diffuse easily within the droplet but are in slower exchange with the cytoplasm. Furthermore, liquid droplets are known to contain proteins that are prone to engage in weak protein–RNA or protein–protein interactions as

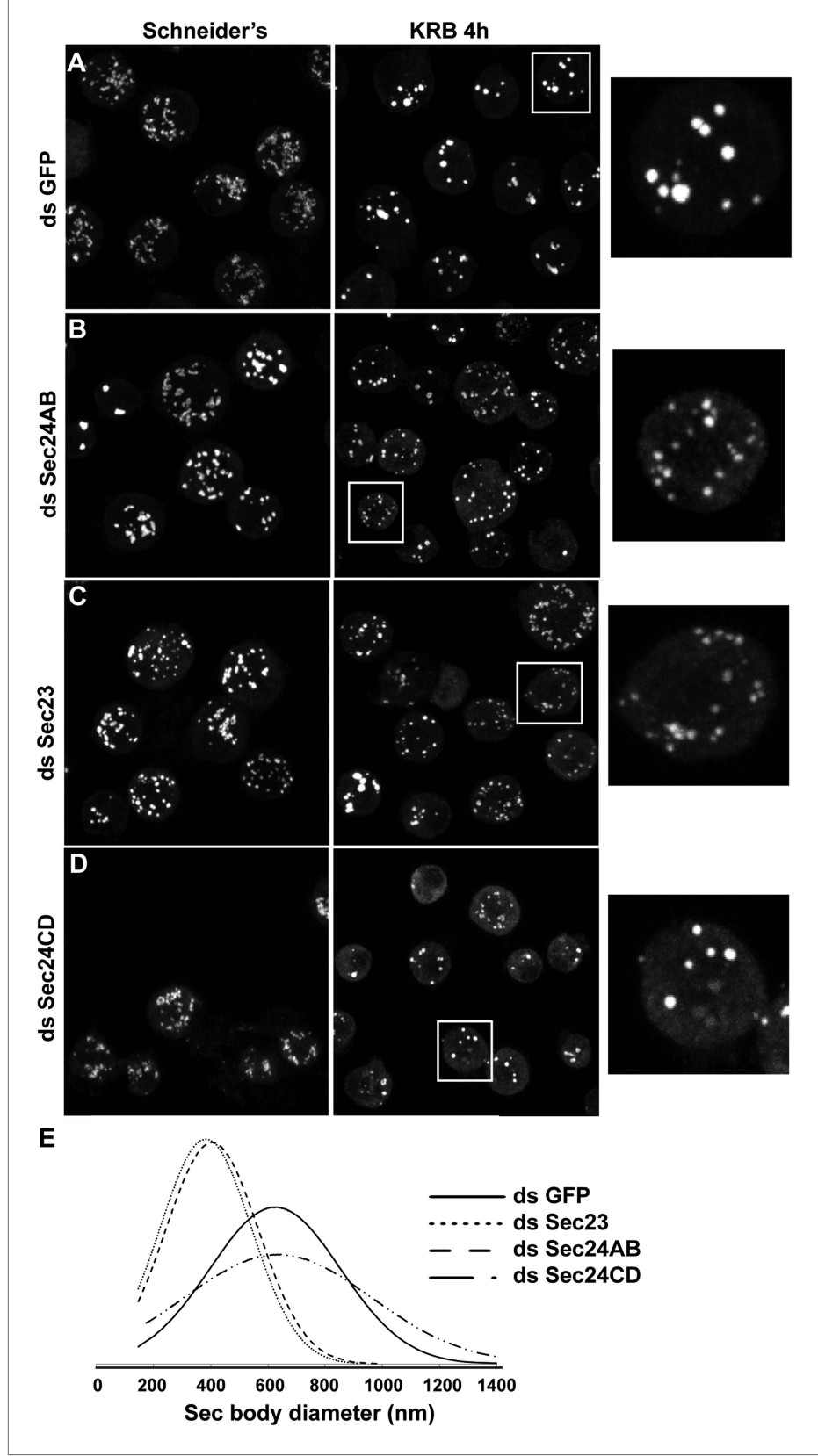

**Figure 5**. Sec23 and Sec24AB are key factors for Sec body formation. (**A**–**D**) IF visualization of Sec16 in mock (**A**), Sec24AB (**B**), Sec23 (**C**), and Sec24CD (**D**) depleted S2 cells in Schneider's and incubated in KRB for 4 hr. Note that
*Figure 5. Continued on next page*

Zacharogianni *et al*. eLife 2014;3:e04132. DOI: 10.7554/eLife.04132

*Figure 5. Continued*
Sec body formation is inhibited upon Sec24AB and Sec23, but not upon Sec24CD depletion. Boxed areas are shown at higher magnification. (**E**) Distribution of Sec body size (shown as frequency of observed Sec body diameter) in mock, Sec24AB, Sec24CD, and Sec23-depleted and starved cells (as in **A**) (dsGFP: 45 cells, 341 Sec bodies; dsSec24AB: 43 cells, 585 Sec bodies; dsSec24CD: 35 cells, 245 Sec bodies; dsSec23: 36 cells, 504 Sec bodies). Note that the Sec body's mean diameter decreases by 1.8-fold upon Sec24AB and Sec23 depletion and that Sec bodies are twice as many. Scale bars: 10 μm.
The following figure supplement is available for figure 5:
**Figure supplement 1**. The smaller structures generated in starved Sec24AB-depleted cells are not Sec bodies.

their components display a high level of low complexity sequences (LCS, defined as regions of low amino-acid diversity) (*Kato et al., 2012*) (*Figure 6—Source data 1*). P-granules in *C. elegans*, nucleoli, but also Stress Granules and P-bodies have been shown to have liquid droplet properties (*Brangwynne et al., 2009*, *2011*; *Hyman and Simons, 2012*). We therefore set out to assess whether Sec bodies are also liquid droplets.

First, we used SEG, a bioinformatics tool that determines the LCS content of proteins recruited to the Sec bodies. Interestingly, we found that Sec16 and the two Sec24 gene products, Sec24AB, Sec24CD (*Figure 6A*) display a significantly higher LCS content when compared to other proteins related to the early secretory pathway and to the entire Drosophila proteome (our analysis, see 'Materials and methods', *Figure 6A'*; *Figure 6—Source data 1*). Sec16 LCSs are situated throughout its sequence with the notable exception of its conserved central domain (CCD, aa 1090–1590) (*Figure 6A*). On the other hand, Sec24AB and Sec24CD LCSs are mostly situated at the N-terminus of the protein sequence in a manner that is partially conserved throughout evolution (*Figure 6—figure supplement 1*). Furthermore, as recently suggested (*Das et al., 2014*), LCSs correspond to a high level of unstructured sequences and we show that it is indeed the case for Sec24AB, Sec24CD, and Sec16 (using HHpred, http://toolkit.tuebingen.mpg.de/hhpred/), *Figure 6—figure supplement 2* and not shown, respectively).

Remarkably, two of the LCS enriched proteins Sec24AB and Sec16 are also required for Sec body assembly, suggesting that this feature might be necessary. However, not all ERES residing and LCS rich proteins are necessary for Sec body formation, as Sec24CD that contains the same amount of LCS, is not.

We then tested whether LCSs were necessary for protein recruitment to Sec bodies and/or sufficient for their formation. We focused on Sec24AB as the LCSs are clustered to the first 415 amino-acids at the N-terminus (Sec24AB LCS) and compared their Sec body recruitment to this of its nonLCS region (aa 416–1184) (*Figure 6B*). LCS-sfGFP is largely recruited to ERES under normal growth conditions although not as efficiently as full-length Sec24AB. Under starvation conditions, it localizes to Sec bodies as full-length Sec24AB and seems to lead to their enlargement. This demonstrates that the LCS rich region of Sec24AB is sufficient to mediate recruitment to Sec bodies. Conversely, the nonLCS region is mostly cytoplasmic and remains largely so upon starvation, although a small pool is recruited to the Sec bodies. This shows that the LCS rich N-terminus region of Sec24AB plays a key role in recruitment of Sec24 to Sec bodies.

We then tested whether the Sec24AB LCS was sufficient to drive Sec body formation. To do so, cells were depleted of endogenous Sec24AB (resulting in the formation of Sec16 positive smaller structures) followed by the expression of Sec24AB LCS. If this is sufficient, we expect that Sec bodies would form. However, although Sec24AB LCS is recruited to the smaller structures, Sec bodies did not significantly form (*Figure 6—figure supplement 3*). This suggests that either the nonLCS region of Sec24AB participates to Sec body formation, even though on its own, it is only slightly recruited, or that one or multiple other factors are involved in driving Sec body formation.

## Sec bodies have FRAP properties compatible with liquid droplets

Second, we assessed whether the FRAP properties of Sec bodies are compatible with liquid droplets, that is, assemblies made through phase separation. When a fraction of such an assembly (GFP marked) is photobleached, the recovery is quick as the molecules within mix instantaneously. However, when entirely photobleached, the recovery is slower as the exchange with the surrounding cytoplasm is not as efficient. We used Sec16-sfGFP and GFP-Sec23 that are efficiently incorporated to Sec bodies.

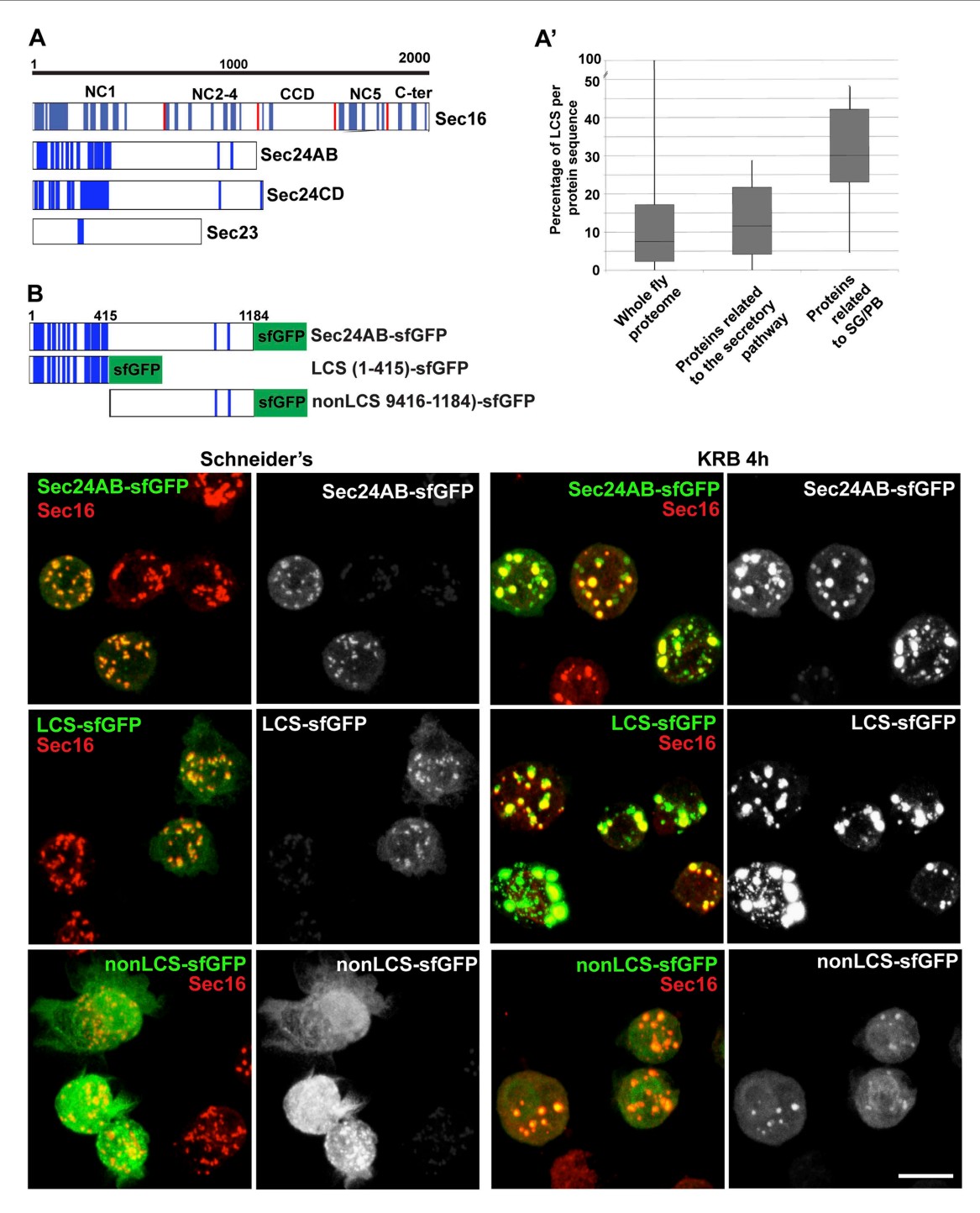

**Figure 6**. Sec body proteins contain low complexity sequences that are necessary for Sec body recruitment. (**A–A'**) Schematic representation of the Low Complexity Sequences (blue bars) in Sec16, Sec24AB, Sec24CD, and Sec23 (A). The red bars mark the boundaries of the Sec16 domains. Genome wide analysis of Low Complexity Sequence (LCS) in the Drosophila proteome, in proteins related to the secretory pathway and proteins related to Stress Granules/P-bodies (**A'**). **B**: IF localization of sfGFP-tagged full-length Sec24B, Sec24AB LCS, and Sec24AB nonLCS in S2 cells in Schneider's and KRB for 4 hr, together with endogenous Sec16 (red). Scale bars: 10 μm.

*Figure 6. Continued on next page*

*Figure 6. Continued*

The following source data and figure supplements are available for figure 6:

**Source data 1**. Table showing the LCS content of proteins related to Stress Granules and P-bodies as well as the early secretory pathway (related to *Figure 6A,A'*).

**Figure supplement 1**. LCS content of Sec24 in different species.

**Figure supplement 2**. Secondary structure prediction of Drosophila Sec24AB using HH pred.

**Figure supplement 3**. Sec24AB LCS is not sufficient to drive Sec body formation.

When Sec bodies are partially bleached, the recovery is very fast for both GFP-Sec23 and Sec16-sfGFP, and the maximum intensity is approximately 50% of the original one. After complete photobleaching, however, Sec bodies recover more slowly and only to 10% of the initial fluorescence intensity, showing an inefficient exchange with the surrounding cytoplasm (*Figure 7A,A'*; *Video 2* and *Video 3*). This is comparable to FRAP properties of Stress Granules that are well-documented liquid droplets. When entirely bleached, Stress Granules recover more than Sec bodies and when partially bleached, they recover slightly less (*Figure 7B,B'*; *Video 4* and *Video 5*. This indicates that the Sec23 and Sec16 diffuse more quickly within Sec bodies than FMR1 in Stress Granules, but that their phase transition barrier is higher.

Taken together, the spherical morphology, specific FRAP properties, and the presence of LCSs are features compatible with Sec bodies being liquid droplets.

## Sec bodies are reversible

Third, we assessed the Sec body reversibility, a key feature of liquid droplets and we tested it using cells that were starved for 4 hr and further incubated in Schneider's. This results in the full recovery of their typical ERES pattern in less than 30 min (*Figure 8A*; *Video 6*; *Figure 8—figure supplement 1*; *Figure 8E*). This convincingly shows that Sec bodies are not terminal aggregates. Furthermore, these ERES are functional as they support efficient transport in the secretory pathway (*Figure 8B*) to allow proliferation (*Figure 9B*, solid dark blue line).

Overall, although we have not been able to determine with certainty whether Sec bodies contain RNAs as all liquid droplets so far characterized do, we propose that amino-acid starvation leads to the formation of a novel stress assembly with liquid droplet features, the Sec bodies.

## Sec bodies act as a reservoir for COPII components and are necessary for cell viability during amino-acid starvation and recovery after stress relief

Remarkably, addition of protein translation inhibitor cycloheximide during the reversal does not affect the ERES re-building (*Figure 8D',E*). This suggests that Sec bodies act as a reservoir for ERES components allowing their re-mobilization upon stress relief to rebuild a functional secretory pathway. Of note, cycloheximide addition during starvation also does not affect Sec body formation (*Figure 8D'*).

To test further whether Sec bodies act as a mechanism for ERES components during starvation preventing their degradation, we first monitored the level of ERES components during starvation. Remarkably, amino-acid starvation leads to an increased level of Sec16, Sec23, and Sec31 (*Figure 9A*; *Figure 9A'*, compare lanes 1 and 2). We can rule out that this is due to an increase in protein translation during starvation as it is efficiently inhibited after 20 min (not shown). It is therefore likely that Sec body formation leads to a stabilization of the ERES components and therefore protect them against degradation.

We then asked whether this protection is inhibited when Sec bodies do not form. Upon Sec16, Sec23 and Sec 24AB depletion, we observed two things: the first is that under normal growth conditions, the level of Sec31 is higher than in mock-depleted cells (*Figure 9A*, compare lanes 3, 5 and 7 to lane 1). The second is that in depleted cells, this level is not maintained upon starvation. Sec31 level is decreased instead of being stabilized as in mock-depleted cells. Sec23 behaves similarly to Sec31. In Sec24AB-depleted cells, however, its level is reduced even in cells incubated in full medium,

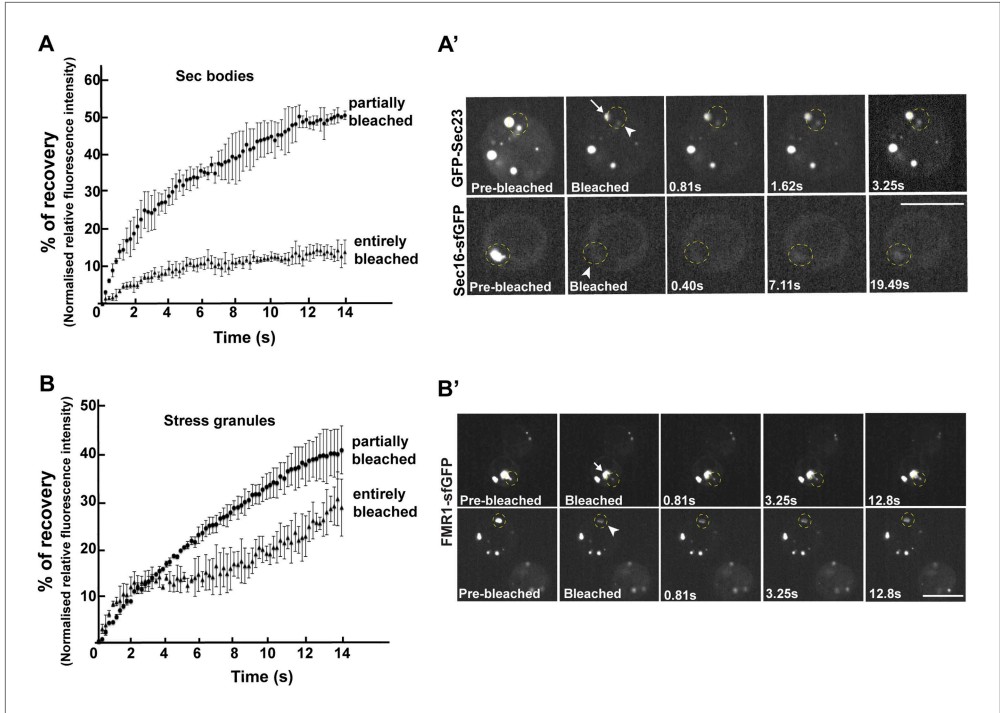

**Figure 7**. Sec bodies have FRAP properties consistent with liquid droplets. (**A–A'**) Percentage of fluorescence recovery after photobleaching (FRAP) over time of individual Sec bodies, marked by GFP-Sec23 (**Video 2**), and ΔNC1-Sec16-sfGFP (**Video 3**) in S2 cells incubated in KRB for 4 hr. The triangles in B show the FRAP of Sec bodies that have been entirely bleached (n = 3, arrowheads in **B'**). The circles show the FRAP of Sec bodies (n = 3, arrows in **B'**) that have been partially bleached. The dashed circles in **B'** indicate the Sec bodies that have been entirely bleached and assessed in stills taken from **Video 2 and 3**. (**B–B'**) Percentage of fluorescence recovery after photobleaching (FRAP) over time of individual Stress Granules marked by FMR1-sfGFP (**Video 4 and 5**) in S2 cells incubated in KRB for 4 hr. The triangles in C show the FRAP of Stress Granules that have been entirely bleached (n = 3, arrowheads in **C'**). The circles show the FRAP of Stress Granules (n = 3, arrows in **B'**) that have been partially bleached. The dashed circles in **B'** indicate the Stress Granules that have been bleached and assessed in stills taken from **Video 4 and 5**. Scale bars: 10 μm.

suggesting that Sec23 turnover depends on the presence of Sec24AB. Altogether, this experiment suggests that starvation leads to ERES component stabilization that is inhibited when Sec bodies do not form. This supports the notion that Sec bodies act as a reservoir for ERES components to rebuild a functional secretory pathway upon stress relief.

In this context, we investigated the relevance of Sec body formation in cell survival upon starvation and fitness upon stress relief. To do so, we exploited the fact that only one of the two Sec24 proteins is required for Sec body formation. Indeed, Sec bodies do not form in starved Sec24AB-depleted cells, whereas they do in Sec24CD-depleted cells (see **Figure 5**). As mentioned above, depletion of either of the Sec24 isoforms is effective but does not affect proliferation very much when cells are grown in normal medium (**Figure 9B**, dashed lines and see 'Materials and methods'). However, upon amino-acid starvation, the number of Sec24AB-depleted cells decreases twice as much as the mock- (ds GFP) and Sec24CD-depleted cells (**Figure 9B**, compare solid green to light blue, violet lines). As expected, the double depleted Sec24AB and CD cells also decline more quickly (**Figure 9B**, solid orange line). When reverted to full medium, control and Sec24CD-depleted cells started to proliferate again, whereas the number of Sec24AB-depleted cells continues to diminish (**Figure 9B**).

As the absence of Sec bodies leads to a decreased stabilization of COPII components, one reason behind the cell lethality upon re-feeding could be the inefficient protein transport through the secretory pathway. To test this, we monitored the transport of Delta to the plasma membrane of starved mock-, and Sec24AB-depleted cells upon re-feeding for 90 min (**Figure 9C**). Mock-depleted cells efficiently

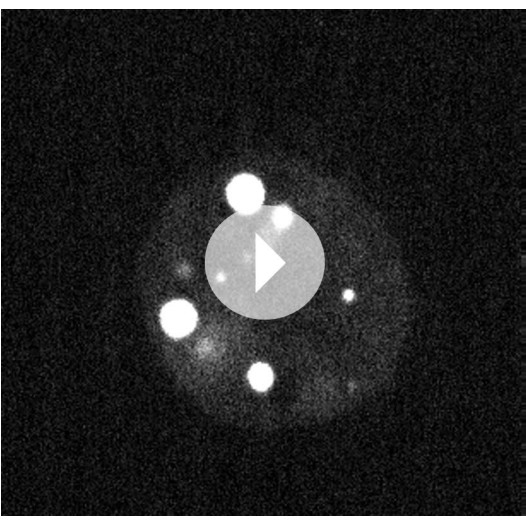

**Video 2**. FRAP of one half-bleached Sec body and one fully bleached. FRAP video of two GFP-Sec23 positive Sec bodies, one photobleached partially and one entirely, recorded every 10 ms for at least 20 s, and then every minute. The video is displayed at 7 frame/s (related to *Figure 7B'*).

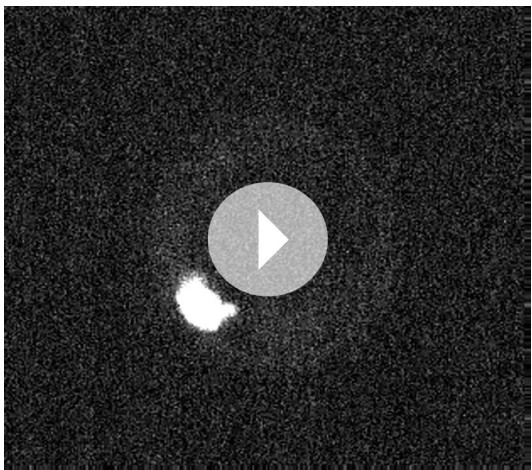

**Video 3**. FRAP of a fully bleached Sec body. FRAP video of one ΔNC1-Sec16-sfGFP positive Sec body entirely bleached, recorded every 10 ms for at least 20 s, and then every minute. The video is displayed at 7 frames/s (related to *Figure 7B'*).

transport Delta whether starved or not (*Figure 9C*; *Figure 8C*). Conversely, the efficiency of transport of Sec24AB-depleted cells after starvation is largely compromised, suggesting that formation of Sec bodies is critical for proper transport resumption. However, this result could simply be due to the depleted Sec24AB whose absence compromises transport even in cells kept in Schneider's.

Nevertheless, Delta transport in Sec24AB-depleted cells kept in Schneider's was found almost as efficient as mock-depleted cells, suggesting that Sec24CD compensates Sec24AB depletion. This indicates that the inhibition of Sec body formation, not the absence of Sec24AB, is detrimental to anterograde transport resumption.

Taken together, these results show that Sec body formation is instrumental to efficient resumption of protein transport through the secretory pathway that contributes to cell survival and growth after re-feeding.

## Discussion

### Sec bodies: a novel stress assembly linked to secretion inhibition

Here, we describe a novel, reversible, and non-membrane bound structure, the Sec body that forms in response to nutrient stress. Sec bodies comprise proteins that in normal growth conditions function as ERES components, including subunits of the COPII coat, namely Sec23, Sec24AB, Sec24CD, and Sec31 as well as Sec16, the upstream ERES organizer. A noticeable exception is the small GTPase Sar1. One reason for this could be that Sar1 is devoid of LCS and this is currently under further investigation.

Interestingly, the components of the other coats were seemingly not incorporated into Sec bodies but also did not form other structures, suggesting that remodeling of the ERES is sufficient to ensure inhibition of protein transport through the secretory pathway. In this regard, Sec body formation constitutes also a novel mechanism for attenuation/inhibition of protein transport through the secretory pathway. Cells can disperse ERES and Golgi components into the cytoplasm, as reported during mitosis the Golgi is fragmented (*Lucocq and Warren, 1987*; *Farhan et al., 2010*; *Zacharogianni et al., 2011*).

The dramatic remodeling of the ERES that we describe here appears to be specific for amino-acid starvation, possibly underlining the acute and severe nature of this particular stress. It is also different from the response to serum starvation that requires ERK7 (*Zacharogianni et al., 2011*). ERK7 appears to be involved to a small extent in the amino-acid starvation response, possibly facilitating the initial dispersion of a fraction of the ERES into the cytoplasm, as when depleted, it prevents Sec body formation upon amino-acid starvation by about 20% (*Zacharogianni et al., 2011*). However, other signaling pathways, yet unidentified, are clearly at stake.

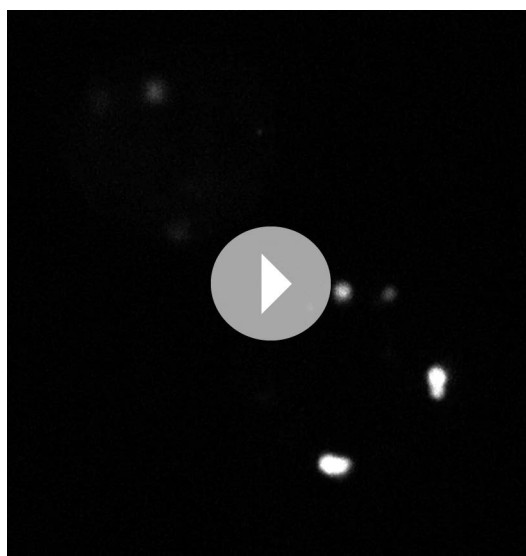

**Video 4**. FRAP of a fully bleached Stress Granule. FRAP video of one FMP1-sfGFP positive Stress Granule entirely bleached, recorded every 10 ms for at least 20 s, and then every minute. The video is displayed at 7 frames/s (related to *Figure 7B'*).

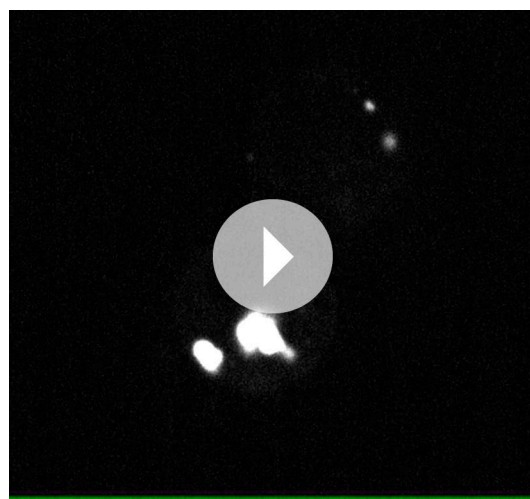

**Video 5**. FRAP of a partially bleached Stress Granule. FRAP video of one FMP1-sfGFP positive Stress Granule partially bleached, recorded every 10 ms for at least 20 s, and then every minute. The video is displayed at 7 frames/s (related to *Figure 7B'*).

Interestingly, one of Sec body components is Sec16, a protein with a localization that is also modulated upon serum starvation. Furthermore, Sec16 is also phosphorylated in response to EGF signaling in human cells (*Farhan et al., 2010*). It could therefore be emerging as one of platform integrating nutrient and growth factors availability.

Sec bodies are novel stress structures and we have shown that they are not autophagosomes (or substrates of autophagy), not lipid droplets and not CUPS as they are devoid of dGRASP that is found quantitatively re-distributed in the cytoplasm upon amino-acid starvation, suggesting a modification involving its lipid anchor or modification of its N-terminus. Sec bodies are also different from large reversible structures containing COPII components that have been described in yeast in a number of specific COPII mutants *Sec12-4* and *Sec16-2* (*Shindiapina and Barlowe, 2010*). These structures are thought to result from an imbalance between cargo incorporation in COPII-coated vesicles and the coat formation, and lowering the cargo load by inhibiting protein translation prevented their appearance. Sec body formation is, however, insensitive to translation inhibition by cycloheximide and is therefore different from these yeast structures. Last, we also show that Sec bodies are distinct from Stress Granules and P-bodies that also form upon amino-acid starvation. Therefore, Sec bodies are novel structures.

## Stress induces the formation of stress assemblies

Formation of mesoscale protein assemblies like Stress Granules, P-bodies or, more recently, Sec bodies is emerging as a general response to stress and especially nutrient stress, and is gaining increasing attention (*Hyman and Brangwynne, 2011*; *Wilson and Gitai, 2013*). For instance, in yeast under nutrient limiting conditions, metabolic enzymes and stress response proteins form reversible foci (*Narayanaswamy et al., 2009*), such as purinosomes containing enzymes of the purine biosynthetic pathway (*An et al., 2008*; *O'Connell et al., 2012*), proteasome storage granules upon glucose restriction (*Laporte et al., 2008*; *Peters et al., 2013*), or, as recently described, glutamine synthetase filaments (*Petrovska et al., 2014*).

In challenging conditions, areas of localized biochemistry in the cytoplasm can be advantageous, as reagents and possibly energy can be focused to these specific areas. The reorganization of the cytoplasm through non-membrane bound protein assemblies could confer this rapid and spatio-temporally defined compartmentalization. In this regard, we have found that Sec bodies confer a fitness advantage to the cells under starvation (see below). However, some stress assemblies (especially cytoplasmic RNP granules) can form dysfunctional RNA–protein assemblies that become irreversible and toxic for the

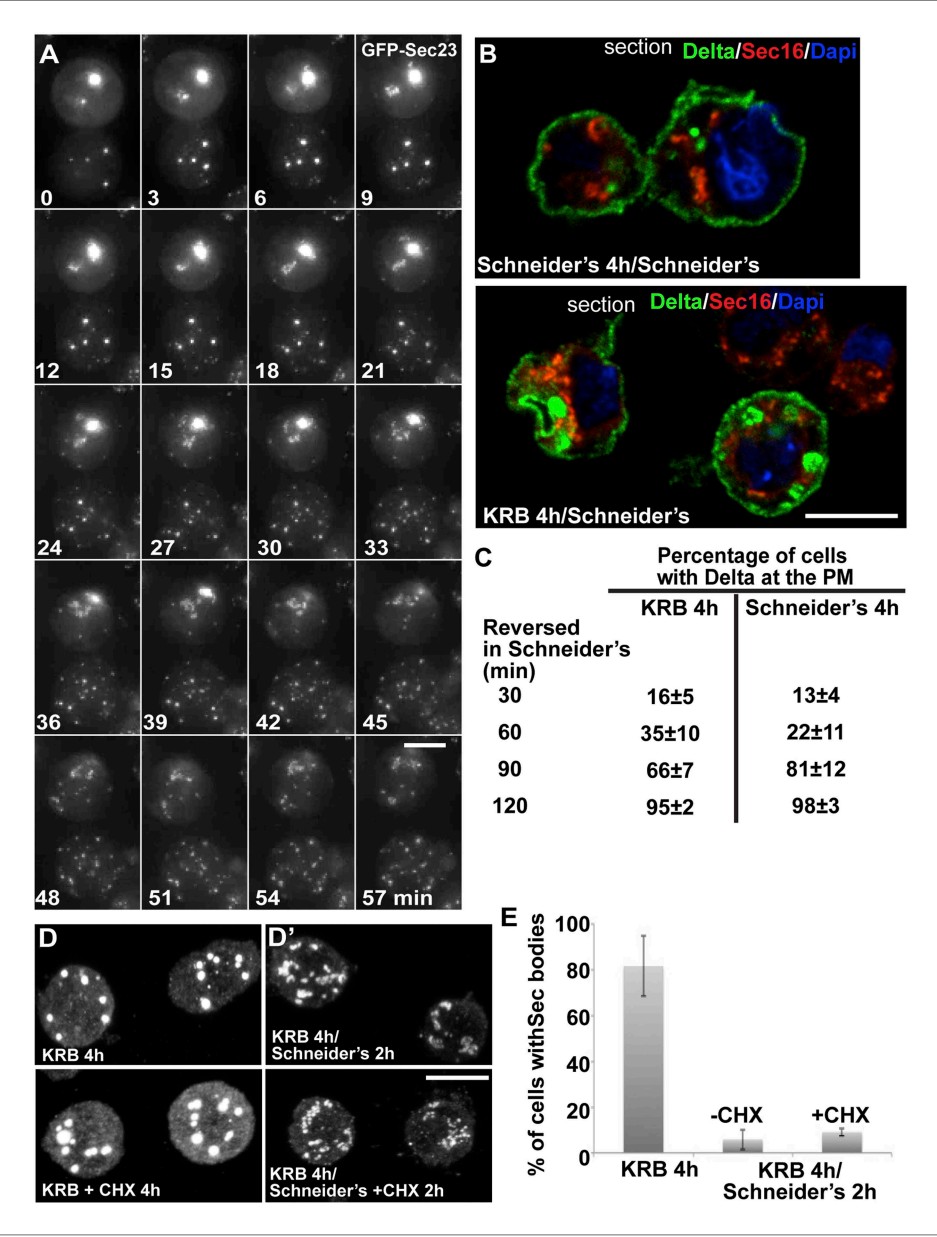

**Figure 8**. Sec bodies are functionally reversible and act as reservoir for ERES components during starvation. (**A**) Stills of a GFP-Sec23 time-lapse video (**Video 6**) of two cells recovering in Schneider's after 4 hr in KRB (t = 0, up to 60 min). Note that Sec bodies are reversed into ERES. (**B**) IF localization of Delta in cells that were starved (KRB) or not (Schneider's) followed further incubation in Schneider's for 2 hr. (**C**) Quantification of the percentage of cells with Delta at the plasma membrane in cells that were either starved (KRB) or not (Schneider's) followed by reversion in Schneider's. Delta was induced for 30, 60, 90, and 120 min while cells were reverted in Schneider's. (**D–D'**) IF visualization of Sec16 in cells starved in KRB supplemented or not with cycloheximide (CHX, **B**), and in starved cells further incubated in Schneider's supplemented or not with CHX (**C'**). Note that neither Sec body formation nor reversal is affected by the presence of CHX. (**E**) Quantification of the Sec body reversal as described in **B'** expressed as the percentage of cells exhibiting Sec bodies.

The following figure supplement is available for figure 8:

**Figure supplement 1**. IF localization of Sec16 in cells recovering in Schneider's for 10–30 min after 4 hr in KRB.

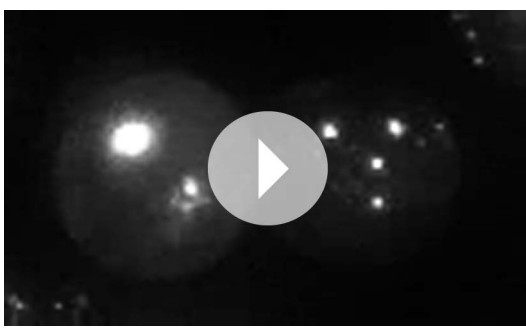

**Video 6**. Sec bodies are reversible. GFP-Sec23 time-lapse video of two cells recovering in Schneider's after 4 hr in KRB (t = 0, up to 66 min). One frame was taken every 3 min and the video is displayed at 7 frame/s (related to **Figure 8A**).

cell. For instance, Stress Granule components have a strong relationship with degenerative diseases, such as ALS and laminopathies (*Ramaswami et al., 2013*). Whether Sec body components could also form such deleterious aggregates remains to be established.

## Some stress assemblies have liquid droplet properties

Some of these mesoscale assemblies have liquid-like properties. These so-called liquid droplets are generally spherical and dynamic and form via phase separation (liquid demixing) of their components from the cytoplasm like a drop of oil in water. Their components display different rates of diffusion within the assembly and in the surrounding cytoplasm (*Hyman and Brangwynne, 2011*). They form via transient and weak protein–protein and protein–RNA interactions mediated by low amino-acid diversity stretches (low complexity sequences, LCS), prone to engage in such interactions. Stress granules and P-bodies have been shown to be liquid droplets, and we show here that Sec bodies exhibit clear liquid droplet features as underlined by their spherical morphology, their reversibility, FRAP properties, and LCS content.

In this regard, the presence of LCSs both in Sec16 and Sec24 is intriguing considering their role in cells under normal growth conditions where they act in sequence with many others to form the COPII coat. How the LCSs are shielded to make proteins competent for their function in COPII coat formation in growing cells remains to be investigated but the interaction with both cargo and Sec23 might be instrumental to their functioning as coat subunits. We show here that the LCS rich domain of Sec24AB is sufficient and necessary for Sec body incorporation upon amino-acid starvation, but not sufficient to induce Sec body formation. This has similarity to Tia1, a key protein necessary for Stress Granule formation that has an LCS-prion like domain that is necessary to form stress granules (*Gilks et al., 2004*).

As mentioned above, the structures that fall in the liquid droplet category have been described to form through weak protein–RNA interactions. Although Sec bodies do not appear to contain RNAs, we propose that they are nonetheless liquid droplets. The absence of RNA might account for the very low and slow recovery we observed after complete photobleaching of whole Sec bodies when compared to Stress Granules that recovers to a higher degree. Shuttling of mRNA in and out of Stress Granules could drive more exchange between the structure and the surrounding cytoplasm, and this probably does not occur in Sec bodies. However, instead of protein–RNA interactions, Sec body components could establish weak protein–protein interactions helped by molecular modifications that could trigger conformational changes and perhaps exposure of their LCS.

## Sec bodies and cell survival

Importantly, one of the key features of a liquid droplet is to be efficiently reversible, and we show that Sec bodies are rapidly and completely reversible upon stress relief. This shows that they act as a reservoir of the ERES components that can be quickly remobilized to re-build a functional organelle, so that protein transport through the secretory pathway can resume once stress is relieved in order to support cell proliferation. Furthermore, Sec bodies have a role in protecting ERES components from degradation during starvation. That strengthens the fact that Sec bodies are neither autophagosomes nor a substrate of autophagy, unlike Stress Granules, which are reported to be cleared by autophagy (*Buchan et al., 2013*). Last, we show that Sec body formation is critical for the cell viability during amino-acid starvation. It suggests a pro-survival mechanism, perhaps through the recruitment and inactivation of pro-apoptotic factors. This remains to be investigated.

Taken together, amino-acid starvation inhibits both protein translation and protein transport through the secretory pathway and, similarly for both processes, results in the concomitant formation of cytoplasmic stress assemblies where key components necessary for cell survival are stored: untranslated mRNAs in Stress Granules and ERES components in Sec bodies.

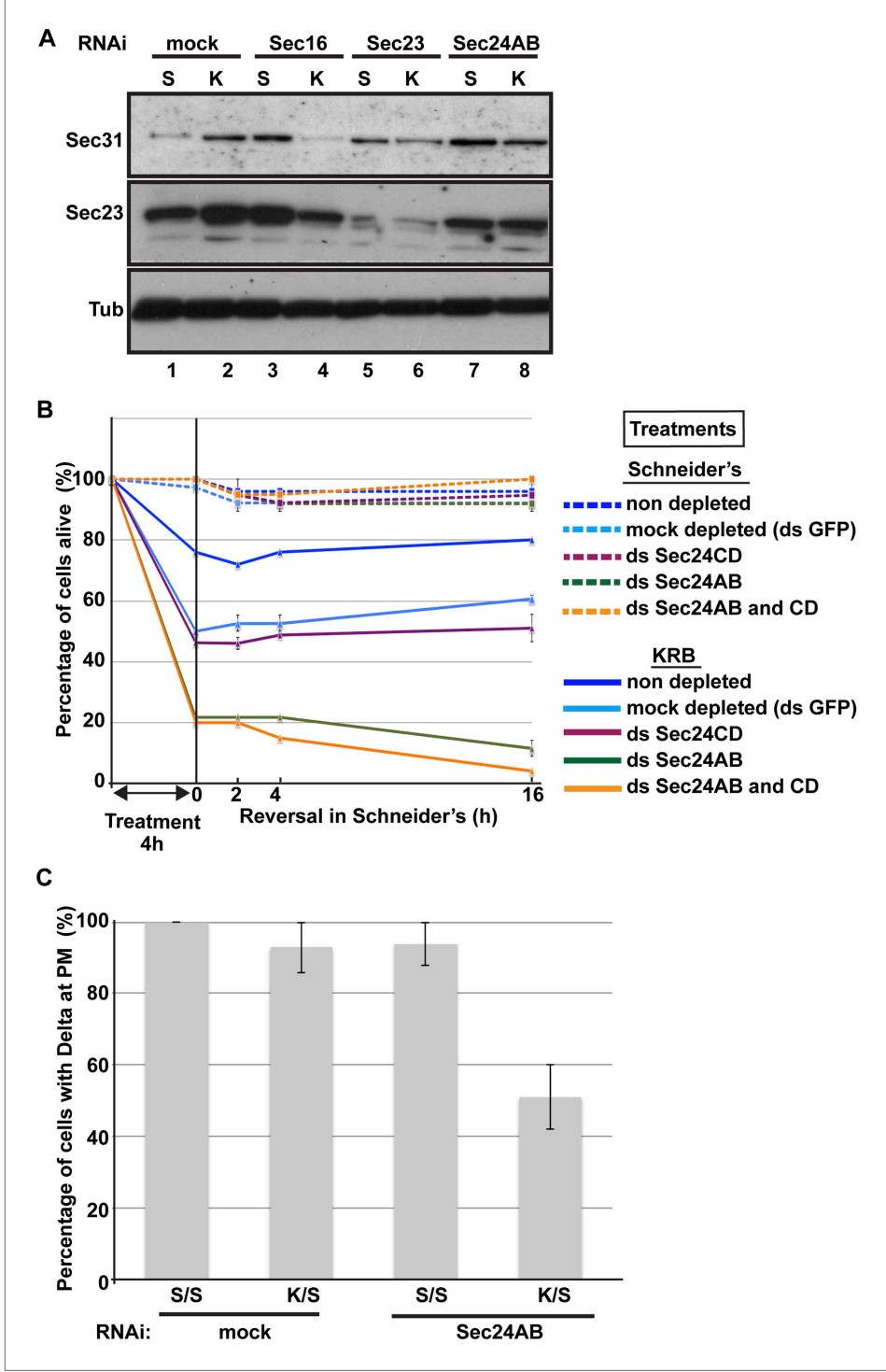

**Figure 9**. Sec body formation is a pro-survival mechanism. (**A**) Western blot of Sec31, Sec23, and tubulin (loading control) of lysates from GFP, Sec16, Sec23, and Sec24AB-depleted cells grown in Schneider's (S) and incubated with KRB for 4 hr (K). (**B**) Graph of cell viability (expressed as percentage of alive cells). The number of starting cells at t = 0, either non- (control, dark blue lines), mock- (dsGFP, light blue lines), Sec24AB (green lines), Sec24CD (violet lines), and double Sec24 AB and CD (orange lines) depleted, is set at 100%. These cells are incubated in Schneider's (dashed lines) and KRB (solid lines) for 4 hr and further incubated in Schneider's up to 16 hr. Note that the mock depletion (dsGFP, light blue dashed lines) is slightly detrimental to cell survival upon amino-acid starvation when compared to non-depleted (control, dark blue dashed line). (**C**) Quantification of the percentage of

*Figure 9. Continued on next page*

*Figure 9. Continued*
cells with Delta at the plasma membrane in mock-, Sec23, and Sec24AB-depleted cells that were either starved (KRB) or not (Schneider's) followed by reversion in Schneider's. Delta was induced for 90 min while cells were reverted in Schneider's. Error bars in B represent standard error of the mean and in C standard deviation.

## Materials and methods

### Cell culture, amino-acid starvation, RNAi, transfection, and drug treatments

Drosophila S2 cells were cultured in Schneider's medium supplemented with 10% insect tested fetal bovine serum (referred to as Schneider's) at 26°C as previously described (*Kondylis and Rabouille, 2003*; *Kondylis et al., 2007*). Amino-acid starvation was carried out by incubating the cells for 4 hr (or otherwise stated) in Krebs Ringers' Bicarbonate buffer (KRB, 10 mM glucose, 0.5 mM magnesium chloride, 4.53 mM potassium chloride, 120.7 mM sodium chloride, 0.7 mM dibasic sodium phosphate, 1.5 mM monobasic sodium phosphate, 15 mM sodium bicarbonate, 5.4 mM calcium chloride) at pH 7.4. We verified that simply adding 10% FBS to the buffer did not prevent the Sec body and Stress Granule formation. Single amino-acids were added at 15 mM (unless otherwise indicated).

Wild-type S2 cells or stably transfected were depleted by RNAi, as previously described (*Kondylis and Rabouille, 2003*; *Kondylis et al., 2007*). Cells were analyzed after incubation with dsRNAs for 5 days (or 4 days in the case of Sar1 depletion). Transient transfections were performed using the Effectene transfection reagent (301425; Qiagen, Germany) according to the manufacturer's instructions. When cells were transfected with pMT constructs, expression was induced 48 hr after transfection with 1 mM $CuSO_4$ for 1.5 hr. The newly synthesized proteins were allowed to localize for 1 hr after $CuSO_4$ washout. When cells were transfected with the pUAS constructs, transfection was done 48 hr prior to the experiment.

Drugs were used at the following concentrations: cycloheximide (10 μM), wortmanin (1 μM), bafilomycin (100 nM), rapamycin (2 μM), and brefeldin A (50 μM). When a drug treatment was followed by starvation, the cells were pretreated for 30 min in Schneider's following starvation in the presence of the drug.

### Antibodies

The following antibodies have been used in these experiments: rabbit polyclonal anti-Sec16 (*Ivan et al., 2008*), 1:800 IF, 1:2000 WB; rabbit polyclonal anti-Sec23 (Pierce PA1-069A), 1:200 IF, 1:1000 WB; rabbit polyclonal anti-Sec31 (*Bentley et al., 2010*) 1:200 IF; mouse monoclonal anti-V5 (life technologies R960), 1:500 IF; mouse monoclonal anti-FMR1 (DSHB supernanant clone 5A11), 1:10 IF; rabbit polyclonal anti-Tral, 1:200 IF; rabbit anti-dGRASP, 1:500 IF (with methanol fixation); mouse monoclonal anti-α-spectrin (DSHB supernatant clone 3A9), 1:20 IF; rat polyclonal anti-eIF4E, 1:200 IF (with methanol fixation); mouse monoclonal anti-Delta, (DHSB clone C594.9B), 1:500 IF.

### DNA constructs and RNAi

The pRmeGFP-Sec23, pRmSar1-eGFP, pMTmini-Sec16 (NC2.3-CCD)-V5, pMTΔNC2.3-V5 constructs were described in the reference *Ivan et al. (2008)*. The pMTΔNC1-ΔCter-Sec16-V5 and the pMTΔNC1-Δ64-Sec16-V5 were described in the reference *Zacharogianni et al. (2011)*. The pMT-Atg5-V5 is a gift from Fulvio Reggiori. The pUASt-GFP-Sec24CD (Sten) and the pUASt-mCherry-Sec24AB were a kind gift from Stefan Luschnig. The Fringe-GFP construct is described in the reference *Kondylis et al. (2007)*.

To generate ΔNC1Sec16sfGFP, sfGFP was amplified using the forward (ggccgcggatggtgagcaa gggcgagga) and reverse (gggtttaaacttacttgtacagctcgtccatg) primers and cloned into PMTV5-B-ΔNC1Sec16 (*Zacharogianni et al., 2011*) using SacII and PmeI restriction sites.

Human Sec16A-V5 was cloned in the pMT-V5-HisB vector using the primers forward (tagccaccgg taccatgcctgggctcgaccga) and reverse (tacggaattcaagttcagcaccaggtgcttcctct) to include the KpnI and EcoRI restriction sites.

To generate FMR1-sfGFP, sfGFP was first amplified using the forward (catgttcgaaatggtgagcaa gggcgag) and reverse (catgaccggtcttgtacagctcgtccatgc) primers containing the restriction sites

for BstBI and AgeI, respectively and cloned into PMT-V5-His to replace the V5 tag, leading to PMT-sfGFP.

To generate super folder GFP (sfGFP) tagged pMT-Sec24AB, pMT-Sec24AB LCS, and pMT-Sec24AB nonLCS, Sec24Ab LCS (1-415 aa) were amplified from cDNA of *Drosophila* S2 cells using the forward (gatggaattccaccatgtcgacttacaat) and reverse (gtcagggccctctggagcagtggttc), and Sec24AB nonLCS (416–1184 aa) regions using forward (cagtgaattcatgctaaacgtggctca) and reverse (gtcagggccctcttttt gcaacacatt) primers. Fragments were cloned into pMT-sfGFP using EcoRI and ApaI restriction sites.

FMR1 cDNA was then amplified from the total cDNA of S2-cells using the forward (catgggtacccacc atggaagatctcctcgtgga) and the reverse (catggaattcaaggacgtgc cattgaccag) primers and inserted in pMT-sfGFP-His using KpnI and EcoRI restriction sites.

The following primers were used to amplify cDNA templates using for RNAi

|  | Forward | Reverse |
| --- | --- | --- |
| Sec24AB | Taatacgactcactatagggggccaaccggtttcaatcag | taatacgactcactatagggaggaggtagctggggttgac |
| ec24CD | Taatacgactcactatagggccctagagtgctccggctat | taatacgactcactatagggcgctctccttcgctgttc, |
| Sec16 | ttaatacgactcactatagggagagccagaggatcagcatc | ttaatacgactcactatagggagagcgatcccacagcagtc, |
| Sec23 | Ttaatacgactcactatagggggtgcaggatatgctcggaat | ttaatacgactcactatagggggtggagctgggattcaatgt, |
| Sar1 | Ttaatacgactcactatagggatgttcacttgggactggttc | ttaatacgactcactatagggagaatctctcgagcccacttcaa |

The DNA fragments were used for in vitro transcription using the T7 Megascript kit (AMBION) to generate the dsRNAs used for RNAi.

The efficiency of Sec24AB and CD depletion was estimated by transfecting mCherry-Sec24AB in Sec24AB-depleted cells and Sec24CD-GFP to Sec24CD-depleted cells and comparing the level of transfection (number of cells expressing the fluorescent protein) to this of non-depleted cells. In a typical experiment, 24.1 ± 2.0% non-depleted cells were transfected with mCherry-Sec24AB vs 2.5 ± 1% in Sec24AB-depleted cells, and 27.3 ± 2.4% non-depleted cells were transfected with GFP-Sec24CD vs 2.6 ± 0.7% in Sec24CD-depleted cells, showing that the 90% of the cells are depleted. The depletion of Sec23 and Sec16 were also 90% (as measured by Western blot, not shown).

## Immunofluorescence (IF) and immuno-electron microscopy (IEM)

S2 cells were plated on glass coverslips, treated, fixed in 4% PFA in PBS for and processed for Immunofluorescence as previously described (*Kondylis and Rabouille, 2003*). Alternatively (as indicated for specific antibodies and dyes), the cells were fixed in ice-cold methanol for 5 min washed with PBS and processed for IF as above. Samples were viewed under a Leica SPE confocal microscope using a 63× lens and 1.5–3× zoom. 17–22 confocal planes are projected to image the whole cells. IEM was performed as described previously (*Kondylis and Rabouille, 2003*; *van Donselaar et al., 2007*).

## Delta transport assay

To monitor Delta transport through the secretory pathway in starved Delta-S2 cells (*Kondylis and Rabouille, 2003*). Delta expression was induced by adding 1 mM CuSO$_4$ to Schneider's medium for 30 min before incubating the cells in KRB or further in Schneider's (control) (*Figure 1A*). To monitor Delta transport upon reversion, cells were incubated 4 hr in Schneider's or KRB. After 4 hr, the media were changed to Schneider's supplemented with 1 mM CuSO$_4$ for 10–120 min (*Figure 8B,C*).

To monitor Delta transport in Delta S2-depleted cells, 0.75 million cells were mock- (dsGFP), Sec24AB-, and Sec23-depleted for 5 days in a 6-well plate cell. Cells were then were split in 2 and plated on glass coverslips in a 12-well plate. They were allowed to attach for 1 hr and the media were changed for either Schneider's or KRB. After 4 hr, the media were changed to Schneider's supplemented with 1 mM CuSO$_4$ for 90 min (*Figure 9C*).

Cells were fixed and processed for IF using Delta antibody. Transport efficiency was calculated as percentage of cells expressing Delta at the plasma membrane over the total number of cells expressing Delta. Between 30 and 60 cells were analyzed per condition.

## Autophagy

S2 cells transiently expressing Drosophila Atg5-V5 and mouse GFP-Atg8 were incubated for increasing length of time in Schneider's supplemented or not with rapamycin and in KRB supplemented

or not with wortmannin and bafilomycin. The percentage of cells showing Atg5-V5 punctae was determined. Experiments were done in triplicate.

## Time-lapse and FRAP

Time lapse of Sec body formation and disassembly was performed on S2 cells stably expressing GFP-Sec23. For Sec body formation, cells were incubated in KRB (t = 0) at 26°C. For Sec body disassembly, cells were starved for 4 hr and further incubated in Schneider's (t = 0) at 26°C. Cells were viewed with a Leica AF7000 Fluorescence microscope. 10 z planes with a z step of 0.7 μm of were recorded every 3 min.

The FRAP experiments were performed on cells expressing GFP-Sec23, ΔNC1-Sec16-sfGFP, or FMR1-sfGFP for 1.5 hr (expression induced with $CuSO_4$) followed by incubation in Schneider's for 1 hr and starvation in KRB for 4 hr. Sec bodies and Stress Granules were entirely or partially (half) photobleached using a 488 nm laser at 100% laser power for 750 msec. FRAP was recorded using a PerkinElmer Ultraview VoX spinning disk microscope with the volocity software. Fluorescence recovery was recorded every 10 ms for the first 14 s after bleaching, and thereafter every 10 s for 2 min.

## Low complexity sequence analysis

The amount of LCS was determined for each protein and isoform annotated in FlyBase release FB2014_02 using SEG (ftp://ftp.ncbi.nih.gov/pub/seg/) (*Wootton and Federhen, 1996*). The LCS content of each protein was tested for enrichment by a hypergeometric test against the whole proteome. For the proteins that have multiple isoforms the longest isoform was chosen for comparison.

## Western blot

Cells were lysed in 50 mM Tris–HCl pH 7.5, 150 mM NaCl, 1% Triton X-100, 50 mM NaF, 1 mM $Na_3VO_4$, 25 mM Na2-β-glycerophosphate supplemented with a protease inhibitors tablet (Roche). The lysate were cleared by centrifugation 4°C for 15 min at 14,000 rpm, and proteins were separated on SDS-PAGE followed by western blot.

## Mammalian cell starvation

HEK-293T, MEFs, and COS7 cells were cultured in standard DMEM and incubated in KRB supplemented with 100 nM bafilomycin for 7 hr (up to 16 hr for Cos cells) to mimic the optimal conditions found for S2 cells.

## Cell survival and fitness upon and after amino-acid starvation

0.75 million cells were non-, mock- (dsGFP), Sec24AB, Sec24CD, and Sec24AB and CD depleted for 5 days in a 6-well plate. They proliferated to reach 2.5, 1.9, 1.8, 1.85, and 1.9 million, respectively and this was set at 100% (t = 0). These cells were then either starved in KRB for 4 hr or further incubated in Schneider's, their number monitored and expressed as a percentage of t = 0. The medium was changed to Schneider's and their proliferation monitored further up to 16 hr. Experiments were performed in triplicates.

## Flies

*Oregon R+* virgin females were fattened on standard food supplemented with yeast for 3 days. They were subsequently either dissected to harvest the ovaries for the ex vivo treatments (incubation in Schneider's and KRB for 4 hr) or transferred to humidified empty vials for 36 hr before dissection. IF were performed as described in the reference *Giuliani et al. (2014)*.

## Quantification and statistics

Three independent experiments were performed for quantification of the Sec body phenotype as scored by immunofluorescence. At least three fields were analyzed comprising at least 100 cells per condition. Averages and standard deviations reflect variation throughout the experiments. For Sec bodies we considered cells with at least one large, round (>0.5 μm) structure as exhibiting Sec bodies. Cells with smaller round structures and/or haze were considered intermediate. For all measurements p-values were calculated with Excel.

## Sec body diameter

Sec body diameter was measured using the Leica LAS software. At least 35 cells were analyzed per condition, in each of which all fluorescent foci (at least 500) were measured. Distribution curves were made using Excel.

## Acknowledgements

We thank the Rabouille's lab members and Fulvio Reggiori for their input and constructive discussions, and Peter van der Sluijs and Adam Grieve for critically reading the manuscript. We thank Akira Nakamura for Tral and eIF4E antibodies, Stefan Luschnig for the tagged Sec24 constructs, and Andrew Bailey for help with the hypoxia experiment. We thank Anko de Graaff and the Hubrecht Imaging Center for supporting the imaging. The spinning disk microscope is funded by equipment grant 834.11.002 from the Dutch Organization of Scientific Research (NWO). Part of this research is funded by NWO 822-020-016 to CR.

## Additional information

### Funding

| Funder | Grant reference number | Author |
|---|---|---|
| Nederlandse Organisatie voor Wetenschappelijk Onderzoek | 822-020-016 | Angelica Aguilera Gomez |
| Royal Netherlands Academy of Arts and Sciences | | Catherine Rabouille |

The funders had no role in study design, data collection and interpretation, or the decision to submit the work for publication.

### Author contributions

MZ, AAG, JS, CR, Conception and design, Acquisition of data, Analysis and interpretation of data, Drafting or revising the article; TV, Acquisition of data, Analysis and interpretation of data

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
