## [Decision Letter]

Thank you for sending your work entitled “The Sec body, a novel ERES component-driven stress assembly that ensures cell viability during amino-acid starvation” for consideration at *eLife*. Your article has been favorably evaluated by Vivek Malhotra (Senior editor), a Reviewing editor, and 2 reviewers, one of whom is a member of our Board of Reviewing Editors. Reviewer 1, Elizabeth Miller, has agreed to reveal her identity.

The Reviewing editor and the other reviewers discussed their comments before we reached this decision. The reviewer comments have been appended below. As you will see, the reviewers felt that the discovery of a starvation induced Drosophila Sec body that sequesters core COPII components represents a significant advance that will be of interest to a broad cell biological audience. While the characterization of the Sec body is of high quality, the reviewers felt that the functional significance of this structure needs to be addressed more fully before the manuscript is suitable for publication. The reviewers have two suggestions for experiments that would address this concern. Both reviewers suggest an analysis of mutations in the Sec body component Sec24AB that selectively perturbs Sec body formation, such as in the disordered region of postulated to contribute to lipid droplet Sec body properties. Such selective mutations will test the model presented in the manuscript and could be used to more accurately assess Sec body function. In addition and/or alternatively, reviewer 2 suggests that more precise assays should be used to test your model that Sec bodies are required to re-build the secretory pathway during recovery from starvation.

Reviewer #1:

The manuscript by Rabouille and colleagues describes an exciting new point of regulation of the COPII coat, the formation of “Sec bodies” upon amino acid starvation. To my mind, this is a significant discovery that adds insight into the relatively poorly understood mechanisms of regulation of ER export. The authors thoroughly investigate which components localize to large inclusions that accrue upon prolonged nutrient stress, identifying the core COPII coat components, Sec23/Sec24, Sec31 and Sec16 as both constituents of the structures and drivers of their formation. The timing of Sec body accumulation is monitored alongside autophagy events, which highlights an interesting property: Sec bodies appear after autophagy has declined, which is significant since COPII vesicles are now known to contribute directly to autophagosome formation. Hence, the timing of stress-related events would seem to involve a cascade of trafficking redirection followed by trafficking shutdown. This appealing model is supported by bioinformatic analysis of the COPII coat that identifies disordered regions that may contribute to “liquid droplet” properties of the COPII coat proteins themselves. The one experiment that I would suggest is to directly assess how one of these disordered regions contributes to inclusion formation: since Sec24AB is required for Sec body formation, deleting the disordered N-terminus should have a similar effect. This would be an elegant experiment since yeast experiments with N-terminally truncated Sec24 do not impact secretion in general (our unpublished data), supporting a regulatory role for this domain. In other respects, the study is solid, well-controlled and an exciting new discovery.

Reviewer #2:

This study describes a novel stress-induced structure called the Sec body and provides an initial characterization in Drosophila S2 cells. This structure forms in response to nutrient starvation and appears to be composed primarily of COPII components involved in ER exit. The structure is reversible, and is proposed to be an important storage depot that can be used to re-build the early secretory pathway (specifically ER exit sites) during recovery from starvation. The strengths of the study are its careful characterization of this structure by high quality microscopy studies and the exclusion of a wide range of alternative possible structures. The interest in this study is heightened by the appreciation in recent years of non-membrane-bound cellular structures that may be regulated sites of specialized biochemistry or storage. Description of the Sec body should add to this growing trend, and stimulate future work on this topic. The main drawbacks of the study that lessen its overall impact are the limitation of analysis to only Drosophila S2 cells (plus a little analysis in Drosophila tissue/larvae), and the unresolved issue of functional relevance.

Specific points:

1) As noted above, the existence of the Sec body outside of Drosophila is not established as yet, and this would seem to help broaden the impact and level of interest in this study. It is worth discussing the evolutionary conservation of this phenomenon (or if available, results from mammalian cells).

2) The importance of the Sec body is hinted at in the last experiment, but by itself, was not especially compelling. It would substantially strengthen the paper if biological relevance were established with either a more specific perturbation and/or a more specific assay for ER exit (as opposed to cell death). The concern is that the cell death outcome may not be causally related to the absence of the Sec body; rather, the Sec24AB knockdown could independently affect both Sec body formation and other cellular processes, the latter of which ultimately cause reduced viability. Below are some suggestions that might help improve this important last part of the paper. I'm not suggesting that all three are needed, but any one would strengthen the case for physiologic relevance/function of the Sec body.

a. The ideal experiment would involve a manipulation that disrupts Sec body formation without affecting other cellular processes. I'm sure the authors have considered this extensively, but is there a mutant of Sec24AB, perhaps in the LCS, that would preclude Sec body formation without affecting its function at ERES? Such a mutant that dissociates these two functions would be well investigating as it would allow Sec body function to be studied with higher precision.

b. An alternative strategy to more specifically probe Sec body function would be to use a more precise assay. If the Sec body is needed to re-establish the early secretory pathway, then secretion should be diminished upon recovery in Sec24AB knockdown cells relative to the situation before starvation, whereas secretion rates should be the same before versus after in Sec24CD knockdown cells (or control cells, where Sec body formation is normal).

c. One could see whether, in the Sec24AB knockdown cells, starvation induces degradation of other Sec body components. In other words, is one purpose of the Sec body to protect these key factors from degradation, as speculated?

---

## [Author Response]

Reviewer #1:

*The manuscript by Rabouille and colleagues describes an exciting new point of regulation of the COPII coat, the formation of “Sec bodies” upon amino acid starvation. To my mind, this is a significant discovery that adds insight into the relatively poorly understood mechanisms of regulation of ER export. […] The one experiment that I would suggest is to directly assess how one of these disordered regions contributes to inclusion formation: since Sec24AB is required for Sec body formation, deleting the disordered N-terminus should have a similar effect. This would be an elegant experiment since yeast experiments with N-terminally truncated Sec24 do not impact secretion in general (our unpublished data), supporting a regulatory role for this domain. In other respects, the study is solid, well-controlled and an exciting new discovery*.

We first thank Dr Liz Miller for her very supportive and positive review.

To test the role for LCS, we performed an experiment along the proposed lines. We expressed Sec24AB LCS-sfGFP (aa 1-415) and Sec24AB nonLCS-sfGFP (aa 416-1184) and compared their dynamics/behavior in starved cells to this of full length Sec24AB-sfGFP.

In normal growth conditions, both FL Sec24AB and its LCS localize to ERES, although the LCS is slightly more cytoplasmic. Conversely, the nonLCS is mostly cytoplasmic with a very small pool localized to ERES.

This is not completely in line with what the reviewer has said of yeast Sec24, that is, that removing part of the N-terminal of Sec24 would not do change its properties. Our experiments would suggest that Sec24AB-LCS behaves much more like the full length than the nonLCS (that corresponds to Sec24 N-terminal truncation of 1/3 of the protein). However, we do not know how long the N-terminal truncation of yeast Sec24 was. Second, our experimental set up is different as we express these chimeric proteins on top of the endogenous Sec24AB and this might change the behavior of the LCS. Third, yeast Sec24 is more similar to Drosophila Sec24CD (Figure 6—figure supplement 2) and we assess here Sec24AB. Last, the amount of structured domains in the N-ter of yeast Sec24 seems higher than the Drosophila counterparts, which might confer different properties.

When cells are starved of amino-acids, FL Sec24AB is incorporated to Sec bodies (as shown in Figure 1) and its LCS also, very efficiently (Figure 6). Conversely, the nonLCS remains largely cytoplasmic and is not efficiently recruited to Sec bodies, although it is present in these structures. This shows that as predicted, the LCS region of Sec24AB is necessary and sufficient for its recruitment to Sec bodies (Figure 6).

We then investigated whether Sec24AB LCS is sufficient to drive Sec body formation. To this end, we depleted cells of Sec24AB, expressed Sec24AB LCS in these depleted cells and asked whether it rescues Sec body formation.

As reported, Sec24AB depletion leads to smaller structures that are not Sec bodies (see below). However, expression of Sec24AB-LCS, although it localizes to the resulting smaller structures, does not lead to the typical large Sec body formation. This suggests that Sec24AB LCS does not drive Sec body formation. This is shown in Figure 6—figure supplement 3.

Reviewer #2:

*This study describes a novel stress-induced structure called the Sec body and provides an initial characterization in Drosophila S2 cells. […] The interest in this study is heightened by the appreciation in recent years of non-membrane-bound cellular structures that may be regulated sites of specialized biochemistry or storage. Description of the Sec body should add to this growing trend, and stimulate future work on this topic. The main drawbacks of the study that lessen its overall impact are the limitation of analysis to only Drosophila S2 cells (plus a little analysis in Drosophila tissue/larvae), and the unresolved issue of functional relevance*.

Specific points:

*1) As noted above, the existence of the Sec body outside of Drosophila is not established as yet, and this would seem to help broaden the impact and level of interest in this study. It is worth discussing the evolutionary conservation of this phenomenon (or if available, results from mammalian cells)*.

We have addressed this point in three ways:

First, we starved three mammalian cell lines, immortalized MEFs, COS cells and HEK cells under similar conditions as S2 cells. We definitely see a remodeling of ERES components in starved MEFs when compared to fed cells (Figure 1—figure supplement 4). However, whether this corresponds to Sec bodies is too early to say. HEK cells detach upon starvation and the response of COS cells is complex (and needs to be investigated further).

Second, Sec bodies formation is a combination of (at least) two factors. One factor is that the absence of amino-acids needs to be sensed and signaled. This is perhaps not similar in mammalian and Drosophila cells. The other factor is that ERES components respond to this signaling. Drosophila proteins that are incorporated in Sec bodies might have different properties from their mammalian counterparts.

To test this, we transfected human Sec16A (the long isoform) in S2 cells. In normal growth conditions, hSec16A localizes partially to ERES and it is efficiently incorporated in Sec bodies upon starvation. This suggests that hSec16A has molecular properties compatible with Sec body recruitment (Figure 1—figure supplement 5).

Finally, we would like to stress that although the conservation of the response to amino-acid starvation needs to be demonstrated and is indisputably important, we feel that the results we got on Drosophila are of sufficient interest on their own to justify publication. Many phenomena are only shown in yeast, for instance the formation of filamentous structures enriched in glutamate synthase upon starvation that was recently published in *Elife*. Furthermore, the concept of liquid droplets has first been illustrated in model organisms (with the P granules in *C. elegans*) and has only be applied later to mammalian stress granules.

*2) The importance of the Sec body is hinted at in the last experiment, but by itself, was not especially compelling. It would substantially strengthen the paper if biological relevance were established with either a more specific perturbation and/or a more specific assay for ER exit (as opposed to cell death). The concern is that the cell death outcome may not be causally related to the absence of the Sec body; rather, the Sec24AB knockdown could independently affect both Sec body formation and other cellular processes, the latter of which ultimately cause reduced viability. Below are some suggestions that might help improve this important last part of the paper. I'm not suggesting that all three are needed, but any one would strengthen the case for physiologic relevance/function of the Sec body*.

This is a fair point. In the original manuscript, based on the fact that protein synthesis was not needed for the reversion of Sec bodies to ERES, we have proposed that Sec bodies act as a reservoir for ERES components. This would allow ERES to be re-built immediately upon stress relief and be instrumental to cell viability.

We now show that ERES are indeed re-built within 20 minutes of growth medium addition and that they efficiently sustain anterograde transport of the plasma membrane reporter Delta, as efficiently as in non-starved cells (Figure 8; Figure 8—figure supplement 1).

We have also monitored the protein level of COPII components Sec23 and Sec31, and showed that are stabilized upon starvation (Figure 9, lanes 1 and 2). This illustrates the notion that Sec bodies act to prevent degradation that might occurs in growing conditions (of note, we do not have a mechanism behind this degradation/stabilization at this stage).

To strengthen our conclusion, we have followed two (b and c) of the reviewer’s suggestions:

*a. The ideal experiment would involve a manipulation that disrupts Sec body formation without affecting other cellular processes. I'm sure the authors have considered this extensively, but is there a mutant of Sec24AB, perhaps in the LCS, that would preclude Sec body formation without affecting its function at ERES? Such a mutant that dissociates these two functions would be well investigating as it would allow Sec body function to be studied with higher precision*.

This is a great suggestion, but we do not have such a mutant yet.

*b. An alternative strategy to more specifically probe Sec body function would be to use a more precise assay. If the Sec body is needed to re-establish the early secretory pathway, then secretion should be diminished upon recovery in Sec24AB knockdown cells relative to the situation before starvation, whereas secretion rates should be the same before versus after in Sec24CD knockdown cells (or control cells, where Sec body formation is normal)*.

We agree that Sec body formation could be linked to other cell processes, some of them important for cell viability. But we have monitored transport from depleted cells in which Sec bodies do not form (Sec24AB depleted) that were first starved and reversed to full medium. Upon Sec24AB depletion, transport after the regimen of starvation/growth resumes less efficiently than in mock depleted cells (Figure 9). Of course, it can be argued that we measure the absence of a COPII component necessary for transport.

However, Sec24AB-depleted cells kept in Schneider’s are almost as efficient in Delta anterograde transport as mock-depleted cells. This is probably because Sec24CD compensates for the loss of AB. Therefore, the transport impairment of Sec24AB depleted cells after the regimen of starvation/growth is likely to be due to the absence of Sec body formation.

Taken together, we can state that Sec body formation allows a quicker resumption of transport through the secretory pathway when compared to situations where they do not form. This is likely to be a factor important for the cell viability upon stress relief. Why and how the cell population decreases rapidly during starvation remains to be investigated.

c. One could see whether, in the Sec24AB knockdown cells, starvation induces degradation of other Sec body components. In other words, is one purpose of the Sec body to protect these key factors from degradation, as speculated?

We have performed this experiment and show that upon conditions where Sec bodies do not form (upon Sec23, 24AB and Sec16 depletion), Sec body components are less stable than in cells where Sec bodies do form.

As mentioned above, the stabilization of Sec31 observed in mock-depleted cells (Figure 9, lanes 1 and 2) is not observed upon depletion of Sec16, 23 and 24AB (compare lanes 2, 4, 6 and 8). Sec23 is also not stabilized in starved cells depleted of Sec16 (compare lanes 2 and 4). Altogether, this shows that Sec bodies do act as a protective mechanism for ERES components.

Curiously, the depletion of Sec16, 23 and 24AB seems to stabilize Sec31 in normal growth conditions. Sec23 is also stabilized in fed Sec16 depleted cells (compare lane 1 and 3). It might be due to a reduction of Sec31 and Sec23 turnover in the absence of other COPII components, perhaps suggesting that Sec31 and Sec23 turnover is enhanced by COPII coat formation. Alternatively, the depletion of a specific ERES component might lead to the compensatory upregulation of others.